# ASMR: Activation-Sharing Multi-Resolution Coordinate Networks for Efficient Inference

**Jason Chun Lok Li**[1]*, **Steven Tin Sui Luo**[2]*, **Le Xu**[1], **Ngai Wong**[1]
[1]Department of Electrical and Electronic Engineering, University of Hong Kong
[2]Department of Computer Science, University of Toronto
`jasonlcl@connect.hku.hk`  `stevents.luo@mail.utoronto.ca`
`{xule, nwong}@eee.hku.hk`

## Abstract

Coordinate network or implicit neural representation (INR) is a fast-emerging method for encoding natural signals (such as images and videos) with the benefits of a compact neural representation. While numerous methods have been proposed to increase the encoding capabilities of an INR, an often overlooked aspect is the inference efficiency, usually measured in multiply-accumulate (MAC) count. This is particularly critical in use cases where inference throughput is greatly limited by hardware constraints. To this end, we propose the **A**ctivation-**S**haring **M**ulti-**R**esolution (ASMR) coordinate network that combines multi-resolution coordinate decomposition with hierarchical modulations. Specifically, an ASMR model enables the sharing of activations across grids of the data. This largely decouples its inference cost from its depth which is directly correlated to its reconstruction capability, and renders a near $O(1)$ inference complexity irrespective of the number of layers. Experiments show that ASMR can reduce the MAC of a vanilla SIREN model by up to $500\times$ while achieving an even higher reconstruction quality than its SIREN baseline.

## 1 Introduction

Neural networks have been proven to be very effective at learning representations of various modalities of data such as images, videos, 3D shapes, neural fields, and many more. In particular, Sitzmann et al. (2020); Mildenhall et al. (2021); Park et al. (2019); Li et al. (2024) have demonstrated that simple coordinate networks, taking in a coordinate system and outputting the modality-specific data, exhibit state-of-the-art (SOTA) expressivity as an implicit neural representation (INR). However, while numerous methods have been proposed to improve the encoding capabilities of an INR, an aspect that is often overlooked is the network's cost of inference[1]. A low cost of inference is of particular importance when the inferencing throughput at test time is restricted by hardware constraints.

Currently, hybrid INRs that make use of explicit representations such as Plenoxels (Fridovich-Keil et al., 2022) and Instant-NGP (Müller et al., 2022) are the best at low-cost inference as they remove the need to learn a complex neural network. However, as we will show in Section 4.4, hybrid methods lose the ability to learn a global implicit representation which is required for tasks such as dataset learning. On the other hand, KiloNeRF (Reiser et al., 2021) and MINER (Saragadam et al., 2022) are purely implicit methods that indirectly reduce cost inference through the use of tiny MLPs. Hao et al. (2022) proposed Level-of-Experts (LoE), which uses shared weights instead of completely independent MLPs to further reduce the cost of inference. However, we analyze and show that existing methods that use an ensemble of weights produce undesirable trade-offs between parameter count and inference cost.

It is remarked that the cost of inference of virtually all existing INR architectures is dependent on both the depth and width of the neural network, which are the direct indicators of the network's

---

*Equal contribution

[1]In this paper, we measure the cost of inference by the number of multiply-accumulate (MAC) operations required by the network's weights. We will use MAC and "the cost of inference" interchangeably.

expressivity. While methods that distribute individual MLPs or weights across a grid of the data can effectively reduce the size of the inference network, the significant increase in parameters goes against the original intention of reducing the overall memory footprint. To this end, we argue that a viable way to genuinely reduce the cost of inference is by decoupling the inference cost from the network depth, which could be achieved by amortizing the per-sample inference cost across the entire data instance. To do so, we combine three ideas: (1) **shared activations**; (2) **multi-resolution coordinate decomposition**; and (3) **position-dependent modulations** to formulate "hierarchical modulation", resulting in an **A**ctivation-**S**haring **M**ulti-**R**esolution (ASMR) network.

As shown in Section 3.3, the immediate benefit of activation sharing is the ability to achieve a near $O(1)$ inference cost with respect to network depth and hence reconstruction quality. This permits an ultra-low MAC model with inference cost going even below 2K MAC. This is roughly the cost of inference of a single hidden layer MLP with 32 hidden units, which has an expressivity barely sufficient in representing a low-resolution image. We validate the robustness of our method by fitting a variety of complex signals, and show that the decoupling effect of ASMR does not come at the cost of affecting the model's original expressivity. In particular, ASMR even improves upon the original SIREN model on fitting megapixel images (Section 4.3) and an entire dataset (Section 4.2). Lastly, we highlight the benefit of using ASMR over hybrid representations (Section 4.4). We demonstrate that ASMR permits the learning of global latent structure of signals, enabling it to encode an entire dataset with a single INR, which oftentimes is infeasible for methods employing explicit features.

To summarize, this paper makes four main contributions: **(i)** We propose a novel hierarchical modulation scheme that efficiently incorporates multi-resolution coordinates with minimal parameter increase. **(ii)** We develop the activation-sharing ASMR, the *first* INR to decouple MAC from its depth. This leads to a near $O(1)$ complexity regardless of the number of layers. **(iii)** The proposed ASMR achieves better quality reconstruction with $500\times$ fewer MAC than a vanilla SIREN in high-resolution image fitting tasks. **(iv)** ASMR remains purely implicit, allowing it to handle tasks that require global latent structure, such as meta-learning.

## 2 RELATED WORK

**Multi-resolution INR** Multi-resolution representation of signals has always been an efficient way to learn INRs. Training INRs to reconstruct a progressive resolution of the original signal has been shown to greatly reduce training speed and the number of parameters. For instance, Saragadam et al. (2022) utilizes Gaussian/Laplacian pyramids to allow INR learning of the much "simpler" residual signals, while Lindell et al. (2022) and Shekarforoush et al. (2022) constrain the frequency bands learned by each INR layer in a coarse-to-fine manner for better interpretability. Instead of training the INR in a multi-resolution manner, some methods have found benefits by simply using multi-resolution coordinate inputs to an INR. Along this line, Müller et al. (2022); Takikawa et al. (2021); Dou et al. (2023) belong to a family of hybrid explicit representation INRs that learn a grid of embeddings at multiple levels of resolution, and concatenate them when feeding into an INR. Lastly, multi-resolution coordinates can also be used to partition the input space into grids, where each grid has its specific tiny MLP (Reiser et al., 2021) or weight layer (Hao et al., 2022).

**Modulated INR** Perez et al. (2018) proposes to modulate a network's activation with a simple affine transformation. Although it is designed to enhance visual reasoning abilities, this technique has proven valuable in the context of INRs. For instance, Chan et al. (2021) injects modulations to a generative adversarial network (GAN) to generate samples. Skorokhodov et al. (2021) also utilizes modulation in generative tasks. They employ low-rank modulation which uses parameters predicted by the hypernetwork as inputs. For minimal overhead, our approach considers bias-only modulations instead. Our method of hierarchical modulations takes inspiration from Dupont et al. (2022a;b); Bauer et al. (2023), which proposes to store data in the form of a data-specific bias vector that serves as the modulator of a SIREN model meta-learned on the entire dataset. In our case, instead of storing an explicit modulation for the data instance, we generate, at inference time, a hierarchy of coordinate-dependent modulations for injecting into each network layer. Later, in Section 4.4, we demonstrate that the data-specific modulation can be used in conjunction with our proposed hierarchical modulation.

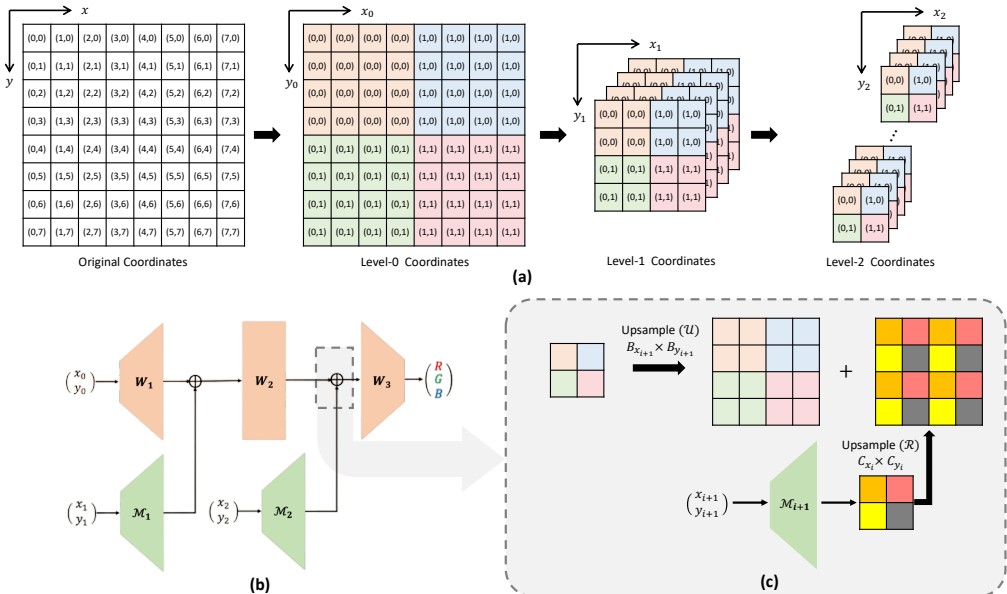

Figure 1: Overall framework for ASMR. **(a) Multi-resolution Coordinates**: The original coordinates are decomposed into multiple hierarchical levels, each with its own set of axes. To illustrate repetitive patterns, the coordinates are folded into a higher-dimensional space. **(b) Hierarchical Modulation**: The number of layers in the model is equal to the number of hierarchical levels. At each level (except level-0), the coordinates are first projected to the hidden dimension via modulators, then added elementwise to the activations of the corresponding layer. **(c) Activation-Sharing Inference**: The MAC-saving activation-sharing inference is performed by utilizing upsampling operations on both modulations and hidden features. Here, $B_{x_i}$ represents the base at level-$i$ along $x$-axis, while $C_{x_i}$ denotes the cumulative product of bases along $x$-axis from hierarchical level-0 to level-$i$ (i.e. $B_{x_i} \times B_{x_{i-1}} \times \ldots \times B_{x_0}$). A uniform base $B_{x_i} = 2$ is used in this example.

**Low-MAC INR**    To our best knowledge, no existing INR architecture can decouple the cost of inference from the depth of the network, where the depth is often the main determining factor of the INR expressivity (Yüce et al., 2022) and hence signal reconstruction quality. In general, existing SOTA INRs that achieve low inference cost either do so through learning an explicit representation (Müller et al., 2022; Takikawa et al., 2021) or learning multiple tiny MLPs (Reiser et al., 2021) to reduce the required size of the INR, or through weight sharing (Hao et al., 2022). Oftentimes, these methods achieve a lower cost of inference at the expense of a largely increased parameter count or decreased reconstruction quality (by reducing hidden dimension). Furthermore, as stated in Reiser et al. (2021), methods that involve training multiple sets of weights often necessitate the use of carefully crafted CUDA kernels to handle non-uniform sampling of the training data. We highlight that ASMR does not suffer from such parallelization constraints due to its nature as a simple modularized addition to a vanilla backbone model. While Skorokhodov et al. (2021) also use activation-sharing like us to cut INR inference costs, they fail to decouple the inference cost from model depth, achieving only minor savings. They utilize multi-scale coordinates, which have progressively increasing resolutions along the network's depth. This significantly increases the modulation cost in the subsequent layers. Contrarily, our ASMR decomposes coordinates considering hierarchical and periodic data patterns, resulting in a complexity reliant solely on the partition basis.

## 3   METHOD

### 3.1   MULTI-RESOLUTION COORDINATES

The idea of multi-resolution coordinates has been introduced in previous literature (Hao et al., 2022; Bauer et al., 2023), but in different contexts. Here, we revisit this idea and provide more intuition underlying it. To begin with, Bauer et al. (2023) interprets coordinate decomposition as a change of

base for the original global coordinates. It is discovered that changing the coordinates into binary representation gives better results in terms of PSNR on image fitting tasks. However, the work does not offer further explanation for this observation. On the other hand, Hao et al. (2022) generalizes this coordinate decomposition motivated by the hierarchical and periodic structure of data. Both explanations refer to the same concept that we refer to as coordinate decomposition.

Expanding on this, we offer an additional perspective that encompasses both explanations. As depicted in Figure 1(a), decomposing global coordinates can be interpreted as an axis partition operation, wherein the axes of the original coordinate system $(x, y)$ are partitioned into shorter axes at multiple levels ($[x_0, x_1, x_2], [y_0, y_1, y_2]$), with $x_i$ and $y_i$ denoting the base of partition. This can be seen as mapping the original coordinates into a higher-dimensional space, enabling subsequent coordinate networks to analyze the repetitive pattern of data. Specifically, given a 1D coordinate[2] $x \in \{0, 1, \ldots, N-1\}$, where $N$ is the input data size (e.g. sequence length of an audio input), $x$ can be decomposed in $L$ hierarchical levels:

$$x \overset{\text{decomp}}{:=} \{x_0, x_1, \ldots, x_{L-1}\} \quad ; \quad x_i \in \{0, 1, \ldots, B_i - 1\}$$

where $B_i$ is base at $i$-th level and $N = \prod_{k=0}^{L-1} B_k$. We call $[B_0, \ldots, B_{L-1}]$ the bases of partition. Let $C_i = \prod_{k=0}^{i} B_k$ be the $i$-th cumulative base and $G_i = \lfloor N/C_i \rfloor$ be the grid size of the partitioned axis at the $i$-th level. The decomposed coordinates are given by $x_i = \lfloor x/G_i \rfloor \mod B_i$. Such a coordinate decomposition creates a hierarchical data representation, starting from a coarse level and progressively refining to a finer level. This computation is only performed once at the beginning of training, making it an efficient way to encode multi-resolution information into the network.

## 3.2 Hierarchical Modulation

Our method encodes information in each resolution level (except level-0) using independent modulators as illustrated in Fig. 1(b), where the number of modulators is equal to $L-1$. The output activations from each modulator are then treated as the bias of the corresponding resolution level. Specifically, an $L$-layer ASMR with hidden dimension $d_i$ in the $i$-th layer is defined as follows:

$$
\begin{aligned}
z_0 &= x_0 \\
z_i &= \sigma(W_i z_{i-1} + b_i + \mathcal{M}_i(x_i)) \qquad i = 1, \ldots, L-1 \\
z_L &= W_L z_{L-1} + b_L
\end{aligned}
\tag{1}
$$

where $z_i \in \mathbb{R}^{d_i}$, $W_i \in \mathbb{R}^{d_i \times d_{i-1}}$, $b_i \in \mathbb{R}^{d_i}$, $\mathcal{M}_i(\cdot) : \mathbb{R}^{d_0} \to \mathbb{R}^{d_i}$ denote the activations, weight matrix, bias and modulator at the $i$-th layer, respectively. $\sigma(x)$ represents the nonlinear activation. In this paper, we employ SIREN (Sitzmann et al., 2020) as our backbone where $\sigma(x) = \sin(\omega_0 x)$ and $\omega_0 \in \mathbb{R}^+$ is a predefined positive scalar to control the frequency of the model.

Prior work (Chan et al., 2021; Dupont et al., 2022a;b) applies modulations to INRs in a global manner, where a low-dimensional latent vector is mapped to a set of global modulations. These modulations are then applied to hidden activations through scales and shifts. Herein we adopt a similar approach to Dupont et al. (2022b), where modulation is applied as a bias vector. However, instead of storing a set of global modulations per INR, we generate "local" modulations by utilizing a position-dependent modulator whereby independent modulators are employed for each resolution level. In particular, the modulator is taken in its simplest form as a projection matrix given by $\mathcal{M}_i(\cdot) := \omega_0 W_{\mathcal{M}_i} \in \mathbb{R}^{d_0 \times d_i}$, where $\omega_0$ is set to be the same as the SIREN backbone and $W_{\mathcal{M}_i} \sim U(-\sqrt{1/d_0}, \sqrt{1/d_0})$. The modulator can be interpreted as storing the eigen-modulations for each hierarchical level. This hierarchical modulation technique serves as a simple but effective way to utilize multi-resolution decomposed coordinates. It improves reconstruction fidelity with only a negligible increase in parameter count.

## 3.3 Activation-Sharing Inference

The fusion of multi-resolution coordinates and hierarchical modulation permits the sharing of activations across data points. In particular, instead of inferring all $N$ data points on every layer,

---

[2]For the ease of illustration, we assume 1D data. However, the idea is easily generalized to multi-dimensional data.

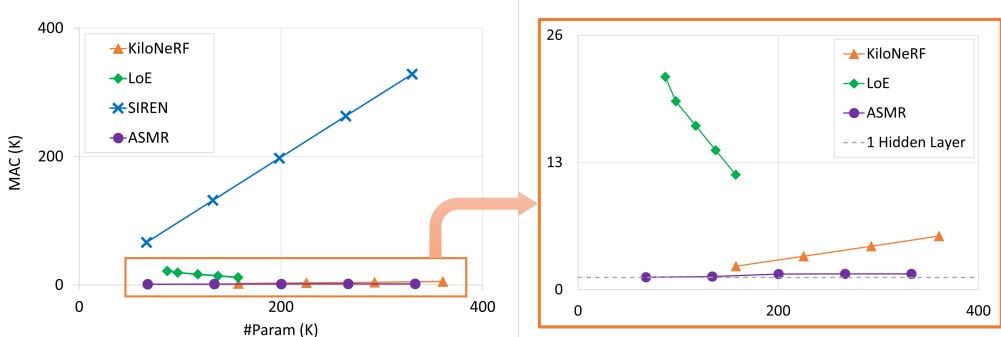

Figure 2: MAC-#Params curves of SIREN, ASMR, KiloNeRF and LoE. We highlight that ASMR reduces the MAC of SIREN models of width 256 by $50 \sim 200\times$ to near the theoretical limit of a single-layer MLP with 32 hidden units (red dotted line).

each hidden layer and modulator only has to infer on $C_i$ data points and $B_i$ grids, respectively, as illustrated in Fig. 1(c). By adopting activation-sharing inference, Equation 1 could be rewritten as:

$$z_i = \sigma(\mathcal{U}(W_i z_{i-1} + b_i, B_i) + \mathcal{R}(\mathcal{M}_i(x_i), C_{i-1})) \tag{2}$$

here, $\mathcal{U}(x, r)$ and $\mathcal{R}(x, r)$ represent the nearest neighbor and tile replication upsampling of the input $x$, performed $r$ times along the corresponding axis, respectively. Such upsampling operations can be easily implemented using `einops` (Rogozhnikov, 2022). This amortizes the inference cost across the entire data, making the per-sample inference cost of ASMR dependent only on the width of the model (i.e. number of hidden units) and independent of the depth, which is contrary and superior to other common INR architecture families such as vanilla SIREN, LoE, and KiloNeRF. This is particularly important because the number of layers directly correlates to the reconstruction quality.

**Proposition 1.** *ASMR decouples its inference cost (in terms of MAC) from its depth L, and consequently its corresponding reconstruction quality.*

*Proof.* Suppose the ASMR model has $L$ layers and each layer-$i$ inferences with a MAC of $M_i$. For notational convenience, we assume that the bases of partition at each level of the ASMR model to be the same[3] (i.e., uniform bases), namely, $B$. Since the cumulative product of all bases must be equal to $N$, i.e. the partitioned grid of the highest resolution must be equal to the resolution of the data, one gets $L = \log_B N$.

For any vanilla MLP, one can approximate its inference cost with the equation MAC $= \sum_{i=1}^{L} N M_i$. Fixing the hidden dimension of each layer gives us an approximation of $M_i$ with $M = \max(M_1, M_2, \ldots, M_L)$. Then, we can simplify the expression for MAC into MAC $\leq NLM$ and the per-sample inference cost is MAC$_{\text{sample}} \approx LM$.

For ASMR, layer-$i$ only has to infer $B^i$ times. Hence, the inference cost of ASMR is given by:

$$\text{MAC} = \sum_{i=1}^{L} B^i M_i \leq M \sum_{i=1}^{L} B^i = MB \frac{B^L - 1}{B - 1} \quad \text{(geometric sum)}$$

$$\approx \frac{MB}{B - 1} B^L = \frac{MB}{B - 1} B^{\log_B N} = \frac{B}{B - 1} MN \leq 2MN$$

since $B \geq 2$. This gives us a per-sample inference cost of ASMR of MAC$_{sample} \approx 2M$ which is solely dependent on the hidden dimension (width) of the model.

---

[3]The non-uniform base case will lead to a much more tedious form of expression, but we note that the idea of amortizing per-sample inference cost could be easily generalized.

Table 1 and Figure 2 summarize the MAC-to-parameter count relationships among low-inference cost architectures analytically and empirically, respectively. For KiloNeRF, its inference cost increases with network depth and hence its overall signal-fitting ability. On the other hand for LoE, its multi-resolution weight-sharing approach induces a negative correlation between its parameter count and depth. This is because an increase in network depth must be accompanied by a decrease in the number of weight tiles at a particular layer by a multiplicative factor that depends on the way

| Model | MACs | #Params |
|---|---|---|
| SIREN | $O(L)$ | $O(L)$ |
| KiloNeRF | $O(L)$ | $O(L)$ |
| LoE | $O(L)$ | $O(\log_2 N - L)$ |
| ASMR | $\underline{O(1)}$ | $O(L)$ |

Table 1: Relating parameter count and MACs to network depth $L$ when network width is fixed.

of partition. LoE asserts that the cumulative product of its weight tile dimensions is equal to the data size, and the smallest weight tile size is 2. Hence, the maximum value of $L$ is $\log_2 N$. Note that both LoE and KiloNeRF reduce their inference cost by increasing their grid resolutions which leads to a significant increase in their parameter counts.

## 4 EXPERIMENTS

All codes are implemented using the PyTorch (Paszke et al., 2019) framework. The baselines are adopted from the official codes provided by the authors, except LoE (Hao et al., 2022) which has no publicly available code, and KiloNeRF (Reiser et al., 2021) and Instant-NGP (Müller et al., 2022) which the original implementation could not conveniently accommodate for our use cases.

### 4.1 ULTRA-LOW MAC TRAINING

To test the performance of ASMR under an ultra-low inference cost scenario, we train the models to fit a 512×512 gray-scale Cameraman (Van der Walt et al., 2014) image. We train 2 sets of SIREN models and apply ASMR to show that our method could achieve more than 50× reduction in inference cost while also increasing the model's reconstruction quality. For the 3-layer SIREN, our ASMR uses partition bases of [8, 8, 8], while the 4-layer SIREN uses partition bases of [4, 4, 4, 8]. As the only additional weights are contributed by the single-layer linear modulators, the increase in parameter count is nearly negligible, especially when considering the significant increase in PSNR for a 4-layer SIREN mode from 32.4dB to 37.8dB. It should also be noted that the reduction in inference cost results in a decrease in training time by 46% from 389s to 210s for the 3-layer SIREN and by 45% from 509s to 280s for the 4-layer SIREN.

| Model (#layers) | #Params (K) | MACs (K) | PSNR(dB) |
|---|---|---|---|
| KiloNeRF (3) | 250 | 0.98 | 34.35 |
| LoE (4) | 126 | 2.07 | 33.27 |
| SIREN (3) | 66.8 | 66.3 | 28.62 |
| SIREN (4) | 133 | 132 | 32.37 |
| **ASMR (3)** | 67.8 | 1.29 | 31.44 |
| **ASMR (4)** | 134 | 1.35 | 37.84 |

Table 2: Ultra-low MAC fitting results on Cameraman image.

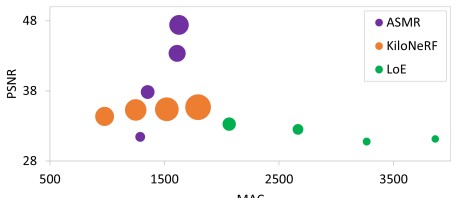

Figure 3: Comparison of MAC-PSNR curves. Circle's area is proportional to #Params.

We also compare ASMR with SOTA low inference cost methods, namely KiloNeRF and LoE, under the ultra-low MAC regime. To simulate a real-life situation where hardware resources are highly restrictive, we limit the parameter counts to less than 150K except for KiloNeRF[4]. It should be noted that even though KiloNeRF and LoE could further reduce MACs by increasing their grid resolutions, our experiments show that they fail to reach $> 30$dB PSNR at a low parameter count. Figure 3 extends the results of KiloNeRF, LoE, and ASMR from Table 2 to include their full MAC-PSNR curves at 3-7 hidden layers. We highlight that the ASMR's near-constant trade-off between inference footprint and reconstruction quality is superior to KiloNeRF's positive linear and LoE's negative linear trade-off.

---

[4]Our experiments in Appendix A show that a KiloNeRF configuration with less than 150K parameters does not achieve $> 30$dB PSNR.

| Method | #Params (K) | MACs (K) | PSNR (dB)↑ | SSIM↑ | Latency (ms) ↓ | | |
| | | | | | Server | Desktop | Raspberry Pi |
|---|---|---|---|---|---|---|---|
| FFN | 132.4 | 131.8 | 31.037±3.233 | 0.841±0.045 | 369.5 | 2859.2 | 8975.4 |
| SIREN | 133.1 | 132.4 | 32.398±3.331 | 0.890±0.037 | 441.4 | 4400.0 | 13476.6 |
| MFN | 134.7 | 133.4 | 27.565±3.132 | 0.782±0.078 | 459.9 | 10291.4 | (Failed) |
| WIRE | 136.8 | 135.9 | 31.266±2.055 | 0.854±0.066 | 2296.5 | (Failed) | (Failed) |
| **ASMR** | 134.7 | **1.85** | **33.087±2.704** | **0.892±0.021** | **143.9** | **957.7** | **3202.0** |

Table 3: Image fitting results (mean ± std.) on the Kodak dataset. The mean ± std. across all images is reported. ASMR achieves the best results in terms of both PSNR and SSIM while having the lowest MACs and latency across various hardware platforms. The reported latency is an average across 10 runs. Processes that are too computationally intensive and result in hanging or being automatically killed due to excessive RAM usage are indicated by (Failed).

## 4.2 NATURAL IMAGE FITTING

To test the robustness of our method on a wide variety of natural images we conduct image fitting on the entire Kodak (Eastman Kodak Company, 1999) dataset. Each image is of either $512 \times 768$ or $768 \times 512$ resolution with RGB channels. The mean and standard deviation across all images are reported. We benchmark ASMR against various baselines, including Fourier Feature Network (FFN) (Tancik et al., 2020), SIREN (Sitzmann et al., 2020), MFN (Fathony et al., 2020), WIRE (Saragadam et al., 2023). Table 3 demonstrates that ASMR surpasses all other baselines in both PSNR and SSIM metrics with a matching number of parameters (approximately 130K), while operating in the ultra-low MAC regime, highlighting the efficiency of our approach. For full implementation details and qualitative results, please refer to Appendix C and Appendix L, respectively.

To demonstrate the improved inference speed of our ASMR method on resource-limited devices, we also measure the rendering latency on various hardware platforms. These platforms range from high-end to low-end CPUs. As shown in Table 3, ASMR consistently speeds up SIREN by $> 3\times$ in terms of latency, and the speed-up becomes more significant when tested on devices with reduced computation power (viz. $> 4\times$ on Raspberry Pi).

## 4.3 OTHER MODALITIES

| Method | #Params (K) | MACs (K) | PSNR (dB)↑ |
|---|---|---|---|
| Instant-NGP | 32.9 | 1.0 | 47.30±3.74 |
| KiloNeRF | 33.4 | 8.16 | 42.20±2.39 |
| SIREN | 33.4 | 33.0 | 46.21±3.30 |
| **ASMR** | 33.8 | 1.0 | **61.66±1.61** |

Table 4: Audio fitting results on the LibriSpeech dataset. The mean ± std. across all samples is reported.

| Method | #Params (M) | MACs (M) | PSNR(dB)↑ | SSIM↑ |
|---|---|---|---|---|
| Instant-NGP | 1.229 | 0.006 | 33.61 | 0.892 |
| KiloNeRF | 1.459 | 0.004 | 27.91 | 0.804 |
| SIREN | 1.054 | 1.054 | 31.66 | 0.825 |
| SIREN-BIG | 4.205 | 4.205 | 35.66 | 0.895 |
| **ASMR** | 1.059 | 0.006 | 33.10 | 0.857 |
| **ASMR-BIG** | 4.215 | 0.008 | 38.73 | 0.938 |

Table 5: Megapixel image fitting on the $8192 \times 8192$ Pluto image.

To showcase the versatility of ASMR across different modalities, we present additional experimental results on audio, megapixel images, video, and 3D shapes.

**Audio.** We evaluate ASMR's performance on an audio-fitting task, using the LibriSpeech (Panayotov et al., 2015) dataset as the benchmark. Our findings suggest that ASMR is particularly effective with low-dimensional data like audio. As shown in Table 4, ASMR, with a matching parameter count of around 33K, significantly outperforms other baselines, including Instant-NGP, KiloNeRF and its SIREN counterpart, with an absolute gain of over 14dB in PSNR, while also exhibiting the lowest MAC of around 1K. Moreover, we also find that ASMR converges much more quickly as compared to the standard SIREN model. For detailed settings, please refer to Appendix D.

**Megapixel Image.** Learning to fit a high-resolution megapixel image is an example where decoupling network parameter count from inference cost is particularly important since oftentimes a significantly larger model is required. Here, we trained our models to fit the $8192 \times 8192$ RGB Pluto (NASA, 2018) image. To achieve >30dB PSNR, a SIREN model must have at least 1M pa-

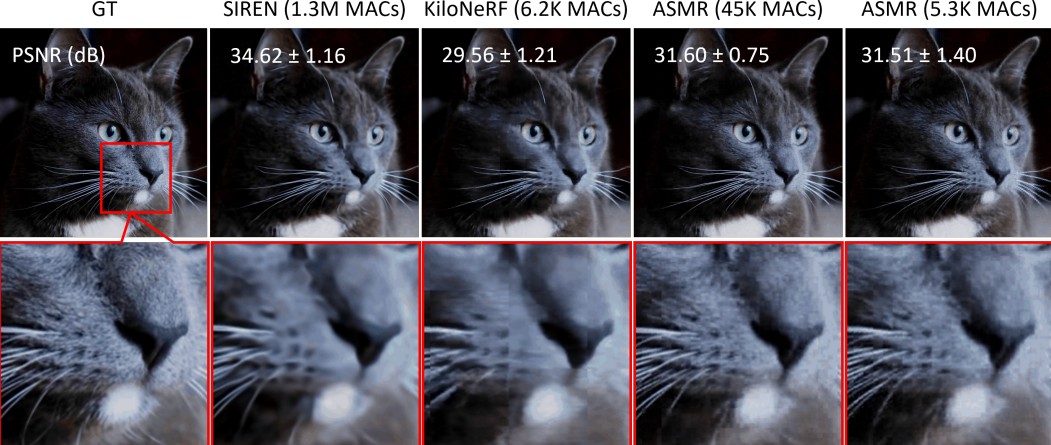

Figure 4: Visual comparison of video fitting results. The mean together with the standard deviation of PSNR across all frames of each model is labeled at the top-left corner of each image. Qualitatively, it is clearly seen that SIREN reconstructs the video with serious smoothing effects while both ASMR models are able to learn the finer details (e.g. whiskers).

rameters (6 layers with 512 hidden dimensions), bringing the cost of inference to more than 1M MACs. However, by applying ASMR to the same SIREN model, we not only could reduce the cost of inference by $\sim 175 \times$ to only 6K MACs, but also increase the reconstruction quality from 31.66dB to 33.10dB PSNR. The reduction in MAC is more significant ($> 500 \times$) when applying ASMR to a larger SIREN with 1024 hidden units (SIREN-BIG), improving the PSNR by more than 3dB. We also benchmark ASMR against the KiloNeRF and Instant-NGP models at similar inference costs and parameter counts. We found that KiloNeRF does not achieve a reconstruction quality comparable to the other models, while Instant-NGP reaches similar levels in both PSNR and SSIM. For detailed settings, please refer to Appendix E.

**Video.** Compared to megapixel images, video data is a complex signal with even more pixels. The additional time dimension also differentiates the nature of video data from that of an image. Here, we follow Sitzmann et al. (2020) and train our models on an RGB cat video (Ehlers, 2019) with 300 frames of 512×512 resolution. The two different configurations of ASMR share the same set of partition bases but have different permutations. This results in varying inference costs, as discussed in Appendix K. A frame where the cat is relatively static is visualized in Figure 4. Despite the original SIREN having a higher PSNR score, we can clearly see that ASMR reconstructs the video at a much higher visual fidelity with fine details encoded. While SIREN has shown significant improvement from the original ReLU MLP with the use of sine activations, we hypothesize that SIREN still suffers from a certain degree of spectral bias due to a fixed scaling factor of $\omega_0 = 30$. On the other hand, KiloNeRF, fitting independent MLPs, exhibits tiling effects where the borders can be seen in the enlarged section. For additional results on the UVG dataset and detailed model configurations, please refer to Appendix G and F respectively.

**3D Shapes.** A direct extension of ASMR to 3D data is demonstrated by training it to fit occupancy grids. One random sample from each category of ShapeNet is selected, amounting to a total of 55 objects, and evaluated with intersection-over-union (IoU). A 10-layer SIREN with 512 hidden units, along with its ASMR counterpart with a basis of partition of 2, are trained over 200 epochs. All hyperparameters remain constant across both models

| Method | MACs (M) | #Params (M) | IoU ↑ |
|---|---|---|---|
| SIREN | 2.100 | 2.10 | 89.27±6.43 |
| ASMR | 0.039 | 2.12 | 89.57±6.72 |

Table 6: 3D shape reconstruction results (mean ± std.) on the ShapeNet dataset.

(see Appendix H for details). The results in Table 6 indicate that ASMR retains the reconstruction quality of SIREN, while significantly reducing inference cost ($> 50 \times$ MAC reduction). However, it is worth noting that ASMR performs significantly worse when trained on continuous 3D signals such as signed distance fields (SDFs) (refer to Appendix M for qualitative results). This suggests that the coordinate decomposition in ASMR has a strong bias towards rasterized data.

## 4.4 META-LEARNED INITIALIZATION

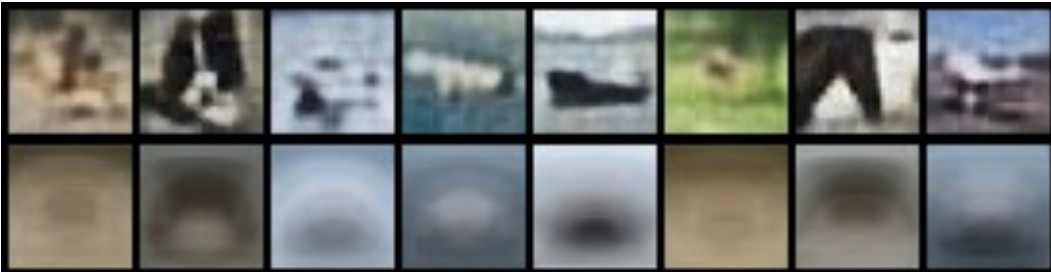

Figure 5: Selected reconstructions from the CIFAR-10 dataset by taking 3 gradient updates from the meta-learned initialization of **(Top)** ASMR **(Bottom)** Instant-NGP.

One distinct advantage that sets ASMR apart from explicit or hybrid INRs, like Instant-NGP (Müller et al., 2022), is its ability to handle tasks that require a global implicit representation, such as generative tasks and tasks involving the use of meta-learning. Here, we follow the setting described in Dupont et al. (2022b) to meta-learn an INR for an entire dataset and learn an instance-specific global latent vector of size 128 to encode each image. Each latent vector is then injected into the INRs as a modulation. Our experiment is conducted on the CIFAR-10 dataset. Fig. 5 displays the selected reconstructions after taking 3 gradient steps from the learned meta-initialization. It is evident that additional instance-specific modulations can be used in conjunction with our hierarchical modulation. Furthermore, it also demonstrates that the multi-resolution hierarchical structure of ASMR can be generalized to effectively encode shared structures across images. In contrast, Instant-NGP fails to learn the shared structure. We argue that this is primarily due to the distortion caused by the highly nonlinear coordinate transformation to an unbounded input space resulting from the hashed encoding in Instant-NGP. For implementation details, please refer to Appendix I.

## 5 LIMITATIONS AND DISCUSSION

Incorporating hierarchical modulations into SIREN introduces a beneficial inductive bias, leading to higher fidelity reconstructions of rasterized data. However, this also causes AMSR to struggle with smooth signals, such as SDFs. The grid-like symmetry bias results in noisy artifacts, especially noticeable when visualizing smooth 3D objects. Furthermore, it also disrupts the clean analytical gradients typically possessed by SIREN. Exploring methods to smoothly extend ASMR to continuous data would be an interesting direction for future research. It is also worth noting that grid-based INRs like Instant-NGP can maintain a fixed MLP's width and depth while enhancing the model's expressivity by enlarging the hash grid embedding. Although ASMR's expressivity is not tied to the MLP's depth, it still depends on the width of the network. Therefore, exploring methods that enable purely implicit INRs to also separate expressivity from the network's width could be valuable.

## 6 CONCLUSION

A novel Activation-Sharing Multi-Resolution (ASMR) coordinate network has been proposed which operates at very low inference costs while maintaining all benefits of an implicit representation. This model combines multi-resolution coordinate decomposition with hierarchical modulations, to allow for the sharing of activations across data grids and the decoupling of inference cost from network depth and reconstruction capability. As a result, ASMR achieves near $O(1)$ inference complexity regardless of the number of layers. We demonstrate that ASMR is the only model, among all baselines, that can work in the ultra-low MAC regime with less than 2K MACs, while achieving a reconstruction result of >30dB PSNR on low-resolution RGB images. Furthermore, ASMR outperforms its SIREN counterpart on large-scale megapixel image fitting tasks with the same parameter count, while significantly reducing MACs by 500×.

ACKNOWLEDGMENTS

This work was supported by the Theme-based Research Scheme (TRS) project T45-701/22-R, and in part by ACCESS – AI Chip Center for Emerging Smart Systems, sponsored by InnoHK funding, Hong Kong SAR.

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

# A    MODEL CONFIGURATIONS IN FIG. 2

The idea of Fig. 2 is to show that our ASMR method allows ultra-low MAC inference and near-constant MAC when increasing the number of layers and hence parameter count. This is a desirable property of an INR as increasing parameter count is a direct indicator of its reconstruction quality.

We compare with two other SOTA INR frameworks that reduce inference cost, namely the Kilo-NeRF family and the LoE family. For all 4 models, we plot their MAC-parameter curve by fixing the hidden dimension and increasing the number of layers progressively. For a fair comparison, we tune the hidden dimension for each model such that the number of parameters is roughly in the same range.

The configurations of all 4 models are summarized in the following table. The configuration of the SIREN model is identical to that of the ASMR model. Note that the "Partition bases $(B_i)$" for LoE is equivalent to weight tile dimensions while that for KiloNeRF is equivalent to grid dimension. Other hyperparameters such as learning rates and position encodings are the same as those presented in the next section.

| Model | hidden_dim | $L$ | Partition bases $(B_i)$ | # Param (K) | MACs (K) |
|---|---|---|---|---|---|
| SIREN | 256 | 7 | N/A | 329 | 323.0 |
| | 256 | 6 | N/A | 263 | 264.2 |
| | 256 | 5 | N/A | 197 | 198.4 |
| | 256 | 4 | N/A | 132 | 132.6 |
| | 256 | 3 | N/A | 66 | 66.8 |
| ASMR | 256 | 7 | [2,2,2,2,2,2,8] | 333 | 1.6 |
| | 256 | 6 | [4,2,2,2,2,8] | 267 | 1.6 |
| | 256 | 5 | [8,2,2,2,8] | 200 | 1.6 |
| | 256 | 4 | [4,4,4,8] | 134 | 1.3 |
| | 256 | 3 | [8,8,8] | 68 | 1.3 |
| LoE | 50 | 10 | [2,2,2,2,2,2,2,2,2] | 87 | 21.8 |
| | 50 | 9 | [4,2,2,2,2,2,2,2] | 98 | 19.3 |
| | 50 | 8 | [4,4,2,2,2,2,2] | 118 | 16.8 |
| | 50 | 7 | [4,4,4,2,2,2] | 138 | 14.3 |
| | 50 | 6 | [4,4,4,4,2] | 158 | 11.8 |
| KiloNeRF | 32 | 6 | [16] | 361 | 5.5 |
| | 32 | 5 | [16] | 293 | 4.4 |
| | 32 | 4 | [16] | 225 | 3.4 |
| | 32 | 3 | [16] | 158 | 2.4 |

Table 7: Model configurations of Figure 2.

# B    IMPLEMENTATION DETAILS OF ULTRA-LOW MAC EXPERIMENT

All models are trained for 10K iterations and configured optimally to have inference costs as low as possible while maintaining a low parameter count and having at least 1 hidden layer.

**Learning Rate** For KiloNeRF, we follow the learning rate stated in Reiser et al. (2021) and use a starting learning rate of 5e-4. For both SIREN and LoE, we did a simple grid search over the set {1e-2, 1e-3, 1e-4} and realized a learning rate of 1e-4 for SIREN and 1e-2 for LoE are the best. For all models, we use the cosine annealing scheduler with a minimum learning rate of 1e-6.

**Position Encoding** Both KiloNeRF and LoE use position encoding as the input to their first layer. We follow the original setting in their papers and set the number of frequencies to 10 for KiloNeRF and 8 for LoE.

The optimal set of hyperparameters obtained in this experiment is also used for the other experiments in this paper.

The following table presents the "boundary" configurations we tested for KiloNeRF and LoE. We show that reducing KiloNeRF's parameter count to a number closer to our SIREN baseline while maintaining a low MAC leads to a sub-30dB PSNR. This explains why we chose to use the configurations in the second row for comparison in Table 2. We also show that further reducing the MACs of LoE while not decreasing parameter count (which makes it unfair to compare with our SIREN (4) and ASMR (4) model) will require reducing the number of layers from 4 to 3. However, this implies that even with a very small number of hidden units (8), the parameter count is still way over our SIREN/ASMR baseline.

| Model | hidden_dim | $L$ | Grid Sizes/ Bases | # Param (K) | MACs (K) | PSNR (dB) |
|---|---|---|---|---|---|---|
| KiloNeRF | 8 | 3 | 16×16 | 108.8 | 0.425 | 23.05 |
| KiloNeRF (in Table 2) | 16 | 3 | 16×16 | 250.1 | 0.977 | 34.30 |
| LoE | 8 | 3 | [[32,16], [32,16]] | 294.9 | 0.361 | 39.47 |
| LoE (in Table 2) | 24 | 4 | [[8,8,8], [8,8,8]] | 126.0 | 1.992 | 36.24 |

Table 8: Boundary configurations and results for KiloNeRF and LoE.

We also include the detailed model configurations of Figure 3, which is an extension of Table 2:

| Model | hidden_dim | $L$ | Partition bases ($B_i$) | # Param (K) | MACs (K) |
|---|---|---|---|---|---|
| ASMR | 256 | 6 | [4,2,2,2,2,8] | 267 | 1.60 |
| | 256 | 5 | [8,2,2,2,8] | 200 | 1.60 |
| | 256 | 4 | [4,4,4,8] | 134 | 1.30 |
| | 256 | 3 | [8,8,8] | 68 | 1.30 |
| LoE | 24 | 7 | [4,4,4,2,2,2] | 38.6 | 3.72 |
| | 24 | 6 | [4,4,4,4,2] | 43.2 | 3.14 |
| | 24 | 5 | [8,4,4,4] | 80.0 | 2.57 |
| | 24 | 4 | [8,8,8] | 126 | 1.99 |
| KiloNeRF | 16 | 6 | [16] | 459 | 1.79 |
| | 16 | 5 | [16] | 389 | 1.52 |
| | 16 | 4 | [16] | 320 | 1.25 |
| | 16 | 3 | [16] | 250 | 0.98 |

Table 9: Model configurations for Figure 3.

## C   IMPLEMENTATION DETAILS OF NATURAL IMAGE FITTING EXPERIMENT

All models are trained for 10K iterations and have around 130K total parameters. FFN is basically ReLU MLP with Gaussian random Fourier features. For MFN, we use the FourierNet instantiation as described in Fathony et al. (2020). Below is the complete set of hyperparameters for all models:

- FFN: lr=1e-3, embedding size=128, $\sigma = 10$, 3 layers, 256 units
- SIREN: lr=1e-4, $\omega_0 = 30$, 4 layers, 256 units
- MFN (FourierNet): lr=1e-2, input scale=256, weight scale=1, 4 layers, 256 units
- WIRE: lr=1e-3, $\omega_0 = 20$, $s_0 = 20$, 4 layers, 212 units
- ASMR: lr=1e-4, $\omega_0 = 30$, bases=([4,4,4,8], [4,4,6,8]), 4 layers, 256 units

Latency results are gathered from various hardware platforms. These include a server-level AMD 7413 2.65GHz 24-Core CPU, a desktop 3.1 GHz Dual-Core Intel Core i5 CPU, and a Quad-core Cortex-A72 (ARM v8) 1.8GHz CPU of Raspberry Pi 4 model B.

## D   IMPLEMENTATION DETAILS OF AUDIO EXPERIMENT

The first 100 samples with a minimum duration of 2 seconds are selected from the test-clean split of the LibriSpeech dataset. All models undergo training for 10K iterations at a learning rate of le-4, using only the initial 2 seconds of each sample. The full set of hyperparameters is detailed below:

- Instant-NGP: levels=7, features per level=2, size of hash map=$2^{16}$, base resolution=125, finest resolution=8000, per level scale=2, 2 layers, 64 units
- KiloNeRF: number of frequencies=12, grid dimensions=[4], 5 layers, 48 units
- SIREN: $\omega_0 = 30$, 4 layers, 128 units
- ASMR: $\omega_0 = 30$, bases=[10, 10, 16, 20], 4 layers, 128 units

## E   IMPLEMENTATION DETAILS OF MEGAPIXEL IMAGE FITTING EXPERIMENT

All models are trained for 500 epochs. We count 1 epoch each time when all pixels in the Pluto image have been sampled once. For KiloNeRF, we randomly sample $2^{22} = 4,194,304$ per training step, while for NGP/SIREN/ASMR, we randomly sample $2^{18} = 262,144$ pixels per training step. We train these four models on a single RTX3090. For SIREN-BIG and ASMR, we sample $2^{20} = 1,048,576$ pixels per step and train the models on two RTX A6000s.

The full set of hyperparameters is detailed below:

- Instant-NGP: levels=16, features per level=2, size of hash map=$2^{16}$, base resolution=16, finest resolution=4096, 3 layers, 64 units
- KiloNeRF: number of frequencies=10, grid dimensions=[16, 16], 6 layers, 32 units
- SIREN: $\omega_0 = 30$, 6 layers, 512 units
- ASMR: $\omega_0 = 30$, bases=[[4,4,4,4,4,8],[4,4,4,4,4,8]], 6 layers, 512 units
- SIREN-BIG: $\omega_0 = 30$, 6 layers, 1024 units
- ASMR-BIG: $\omega_0 = 30$, bases=[[4,4,4,4,2,16],[4,4,4,4,2,16]], 6 layers, 1024 units

In Figure 6, we visualize the reconstructed image at 10 training epochs for all 4 models and realize that ASMR, compared to its SIREN counterpart, can learn the high-fidelity features faster in terms of training steps. This is similar to that of Instant-NGP. On the other hand, KiloNeRF exhibits very blurry tiling effects at such an early stage. Combining the findings from Takikawa et al. (2021) which also uses a set of multi-resolution embeddings, we believe that this is a benefit contributed by the multi-resolution coordinate system.

We also attempted to train a LoE model but the lack of original code made the training for megapixel images prohibitive (days on NVIDIA RTX3090). As random sampling is required for training a

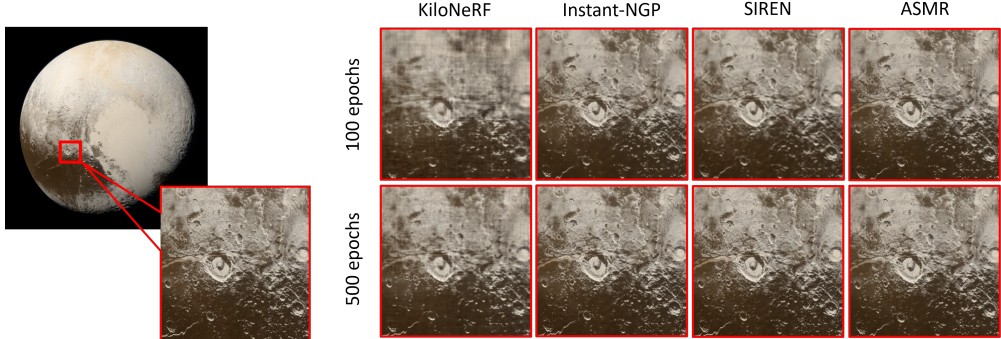

Figure 6: Comparison of reconstruction quality of the Pluto Image at 100 and 500 training epochs.

megapixel image (as well as other large-scale data formats such as video), carefully written CUDA routines become necessary for the parallel training of weights. In order to match the low MACs of our ASMR model while maintaining a similar parameter count, the only possible configuration of LoE is to either have two hidden layers of 16 hidden units and weight tile dimensions of $32 \times 32$, $16 \times 16$, and $16 \times 16$, or a single hidden layer and only 4 hidden units.

## F    IMPLEMENTATION DETAILS OF VIDEO FITTING EXPERIMENT

All models are trained for 200 epochs. We count 1 epoch each time when all pixels in the Cat video have been sampled once. For all models, we randomly sample 153600 pixels per training. We measure the PSNR and SSIM scores per frame and take the average across all 300 frames.

The full set of hyperparameters is detailed below:

- KiloNeRF: number of frequencies=10, grid dimensions=[16,16], 6 layers, 32 units
- SIREN: $\omega_0 = 30$, 7 layers, 512 units
- ASMR-1: $\omega_0 = 30$, bases=[[5,3,1,2,1,2,5], [4,2,2,2,2,2,4], [4,2,2,2,2,2,4]], 7 layers, 512 units
- ASMR-1: $\omega_0 = 30$, bases=[[5,5,3,2,1,1,2], [4,4,2,2,2,2,2], [4,4,2,2,2,2,2]], 7 layers, 512 units

The rest of the configurations of each model and their video-fitting results are presented in the following table.

| Model | # Param (M) | MACs (K) | PSNR (dB) $\pm$ std | SSIM $\pm$ std |
|-------|-------------|----------|---------------------|----------------|
| SIREN | 1.314 | 1313.79 | $34.641 \pm 1.159$ | $0.9079 \pm 0.01062$ |
| KiloNeRF | 1.223 | 6.21 | $29.560 \pm 1.208$ | $0.8289 \pm 0.0228$ |
| ASMR-1 | 1.326 | 5.36 | $31.506 \pm 1.401$ | $0.8536 \pm 0.0340$ |
| ASMR-2 | 1.326 | 44.99 | $31.596 \pm 0.751$ | $0.8189 \pm 0.0122$ |

Table 10: Video fitting results on the Cat video.

## G    ADDITIONAL VIDEO FITTING RESULTS

To further validate ASMR's performance on video data, ASMR is evaluated on the UVG dataset, a common video benchmark consisting of 7 HD videos captured at 120fps. Due to hardware constraints, we downsampled the videos by a factor of 4 to 150 frames with a resolution of $270 \times 480$. This allows a model to achieve good reconstruction quality while fitting into a single NVIDIA 3090 GPU. For the "shakendry" video, however, there are only 75 frames. During each training iteration, we sample 194,400 pixels.

The full set of hyperparameters is detailed below:

- Instant-NGP: levels=17, features per level=2, size of hash map=$2^{16}$, base resolution=[5, 9, 12], finest resolution=[75, 135, 240], 3 layers, 64 units

- KiloNeRF: number of frequencies=10, grid dimensions=[3, 5, 5], 8 layers, 64 units

- SIREN: $\omega_0 = 30$, 10 layers, 512 units

- ASMR: $\omega_0 = 30$, bases=[[5,1,1,5,1,1,3,1,1,2], [5,1,3,1,3,1,3,1,2,1], [5,3,2,2,1,2,1,2,1,2]], 10 layers, 512 units

| Method | #Params (M) | MACs (M) | PSNR(dB)↑ | SSIM↑ |
|---|---|---|---|---|
| Instant-NGP | 1.990 | 0.006 | $29.8 \pm 3.4$ | $0.83 \pm 0.07$ |
| KiloNeRF | 2.194 | 0.025 | $30.3 \pm 4.3$ | $0.83 \pm 0.09$ |
| SIREN | 2.105 | 2.105 | $33.6 \pm 3.6$ | $0.91 \pm 0.04$ |
| **ASMR** | 2.119 | 0.120 | $32.3 \pm 4.2$ | $0.86 \pm 0.07$ |

Table 11: Video fitting results on the UVG dataset.

## H  IMPLEMENTATION DETAILS OF 3D SHAPES EXPERIMENT

We modified the NGLOD  (Takikawa et al., 2021) repository for our occupancy grid experiments. Both SIREN and ASMR are trained for 200 epochs, where each epoch samples 250k points and each training step has a batch size of 4096. Among the sampled points, 60% are sampled randomly, 20% are sampled on the surface, and 20% are sampled near the surface.

The full set of hyperparameters is detailed below:

- SIREN: $\omega_0 = 30$, 10 layers, 512 units

- ASMR: $\omega_0 = 30$, bases=[[2,2,2,2,2,2,2,2,2,2], [2,2,2,2,2,2,2,2,2,2], [2,2,2,2,2,2,2,2,2,2]], 10 layers, 512 units

## I  IMPLEMENTATION DETAILS OF META-LEARNING EXPERIMENT

We use the same setting as in Dupont et al. (2022b), where SGD is used for the inner loop with a learning rate of 1e-2, and the Adam optimizer is used for the outer loop with a learning rate of 3e-6. We set the size of the latent vector to be 128 and allow 3 gradient updates to generate the reconstruction. The latent vector is first mapped to the instance-specific modulation with a size equal to the hidden size multiplied by the number of layers minus 1.

To train ASMR under COIN++ (Dupont et al., 2022b) framework, one can rewrite Equation 1 as

$$z_i = \sigma(W_i z_{i-1} + b_i + \mathcal{M}_i(x_i) + \phi_{i-1}) \qquad i = 1, \dots, L-1 \tag{3}$$

where $\phi = [\phi_0, \phi_1, \dots, \phi_{L-2}]$ is the instance-specific modulation vector to encode each image.

For Instant-NGP, we directly use the hashed encoding implementation provided by `tiny-cuda-nn` (Müller, 2021). The meta-learning code is adapted from the official code released by the author (Dupont et al., 2022b). We use a batch size of 64 during training. The set of hyperparameters used by each model is given:

- Instant-NGP: levels=4, features per level=2, size of hash map=$2^{16}$, base resolution=2, per level scale=2, 5 layers, 512 units

- ASMR: $\omega_0 = 50$, bases=[2,2,2,2,2], 5 layers, 512 units

## J   INDUCTIVE BIAS OF ASMR

The strong performance of ASMR can be attributed to its assumption of a hierarchical, multi-resolution structure in the underlying data. This assumption provides a powerful inductive bias, allowing ASMR to effectively represent the high-frequency components of the data. Fig. 7 compares the reconstructed outputs of ASMR and SIREN during the early training stage on the Cameraman image. It shows that ASMR starts with a predefined hierarchical repeating pattern on the underlying image, while SIREN begins with a relatively low-frequency reconstruction. This inductive bias allows ASMR to converge more quickly in the early iterations as well as achieving better reconstruction quality at the end.

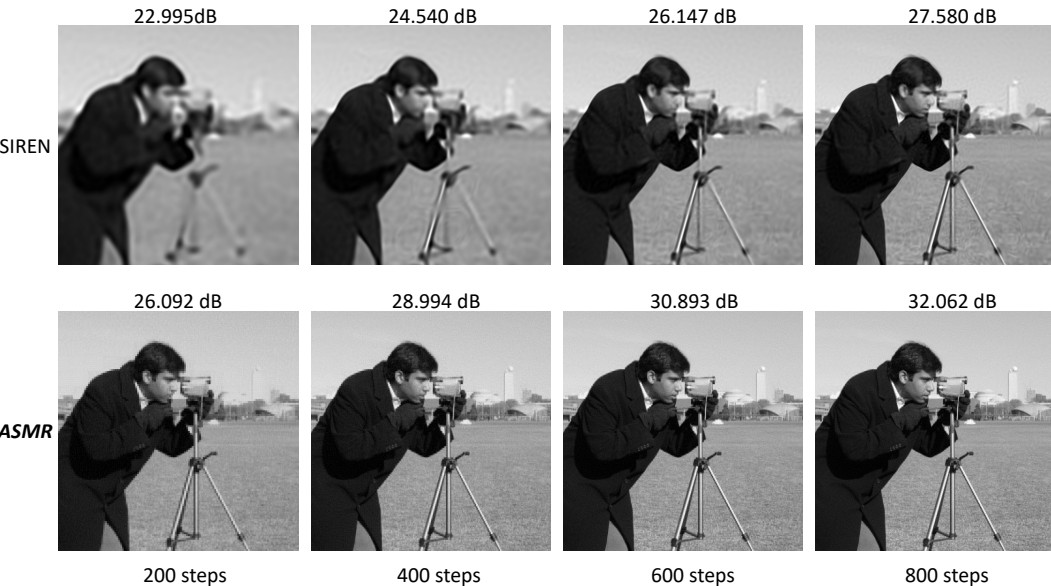

Figure 7: A comparison of inductive bias between ASMR and SIREN, wherein reconstructed images at the early training stage (200, 400, 600, 800 steps) are shown. By imposing a multi-resolution hierarchical structure on the target image, ASMR can achieve faster convergence and accurately represent fine details from the start. In contrast, SIREN initially produces a relatively blurry reconstruction.

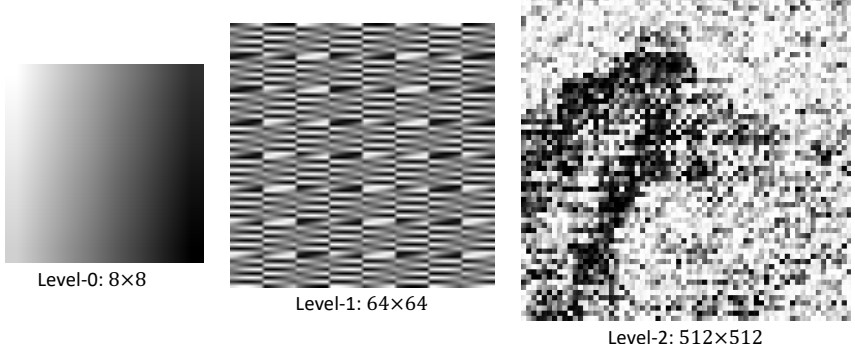

Figure 8: Feature maps obtained from a selected hidden unit of a 3-layer ASMR. It is worth noting that the feature maps shown here are not to scale, and we specify their dimension at the bottom of each feature map.

To provide a better understanding of the underlying activation-sharing mechanism of ASMR, we visualize the hidden activations of ASMR. The ASMR we employ consists of 3 layers with 256 hidden units. In the case of ASMR, we use uniform bases of [8,8,8] for axis partitioning along both the horizontal and vertical dimensions. For brevity and better visualization, we display the activation

of only one selected channel for illustrative purposes. Figure 8 depicts the concept of activation-sharing inference in ASMR. In this approach, a modulation vector is shared among coordinates within the same partitioned grid. Subsequently, a sequence of upsampling operations is performed, leading to a progressively larger feature map in ASMR. Each layer contributes more details, and finer details are repeated more frequently due to the hierarchical structure of decomposed coordinates.

## K   PERMUTATION OF BASES

To provide a good heuristic for choosing the appropriate partition pattern, we conducted a set of experiments on the Cameraman image. In particular, we permute the bases of an ASMR model with 4 layers and 512 units, totaling 134K parameters. Table 12 shows that when more activations are shared at a later stage of the ASMR model using a larger base $B_3$ or equivalently a smaller grid size $G_3$, it reduces the number of MACs without compromising the performance in terms of PSNR and SSIM. This entails that ASMR is insensitive to the permutation of bases, avoiding an exponential increase in hyperparameters to tune during training.

| Partition $[B_0, B_1, B_2, B_3]$ | MACs (K) | PSNR (dB) ↑ | SSIM ↑ |
|:---:|:---:|:---:|:---:|
| $[4, 4, 4, 8]$ | **1.34** | 37.755 | **0.930** |
| $[4, 4, 8, 4]$ | 4.42 | 37.442 | 0.925 |
| $[4, 8, 4, 4]$ | 4.61 | **37.776** | 0.928 |
| $[8, 4, 4, 4]$ | 4.61 | 37.574 | 0.927 |

Table 12: The effect of the bases permutation.

# L    QUALITATIVE RESULTS ON KODAK IMAGE FITTING TASK

In this section, we present some qualitative results on selected images from the natural image fitting task in Section 4.2.

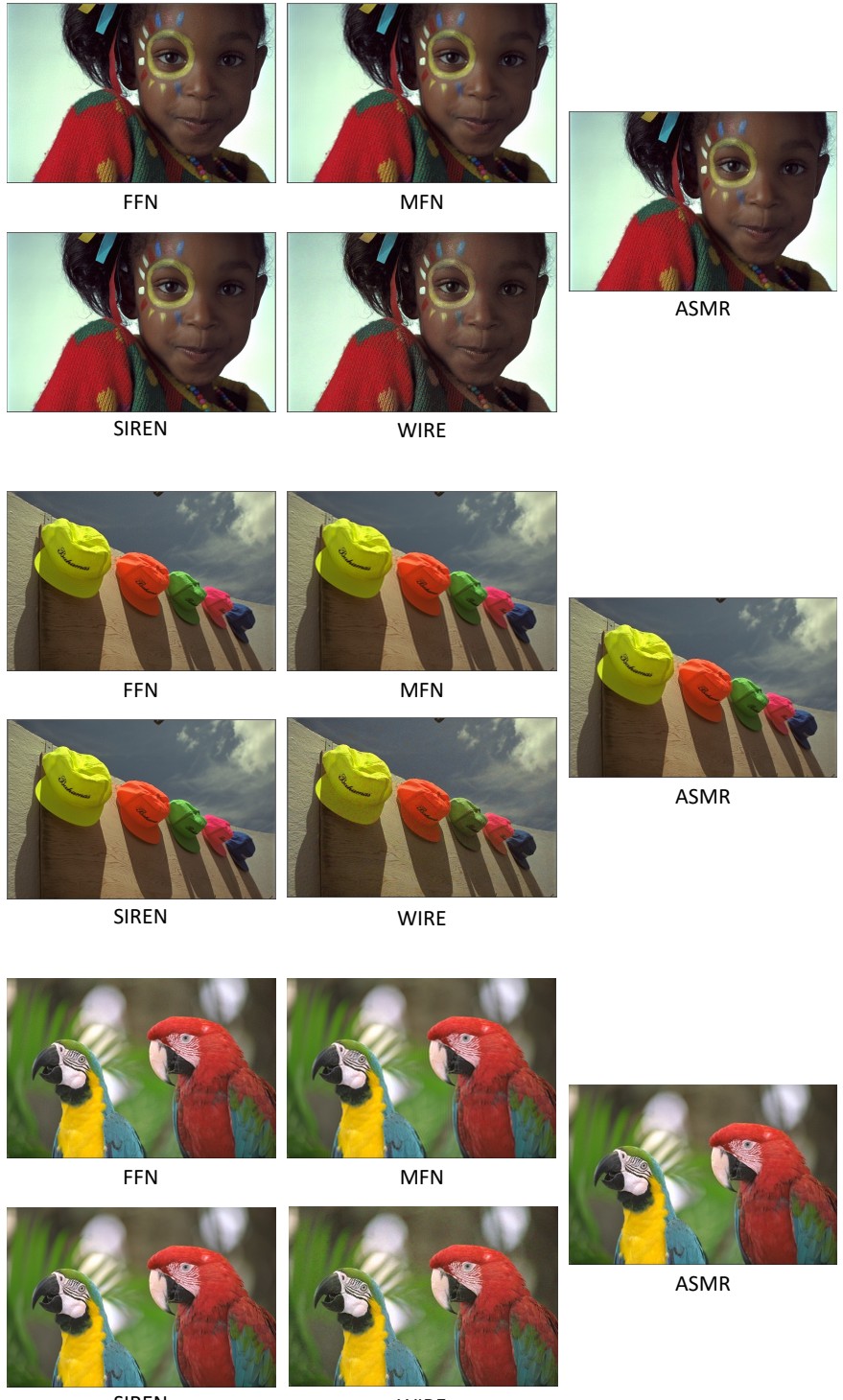

Figure 9: Qualitative results of natural image fitting tasks are shown. The selected images for comparison, from top to bottom, are Kodak03, Kodak15, and Kodak23.

## M    QUALITATIVE RESULTS ON SDF FITTING TASK

In this section, we showcase qualitative results from selected objects in the SDF fitting task. The strong inductive bias of ASMR towards rasterized data hinders its ability to encode continuous signals like SDFs. This deficiency is evident in Figure 10 and Figure 11, where ASMR struggles to capture sharp edges and generates noisy samples.

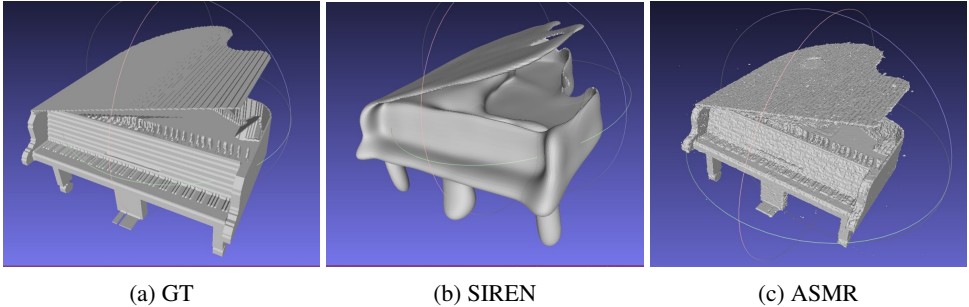

| (a) GT | (b) SIREN | (c) ASMR |

Figure 10: SDF of a piano object.

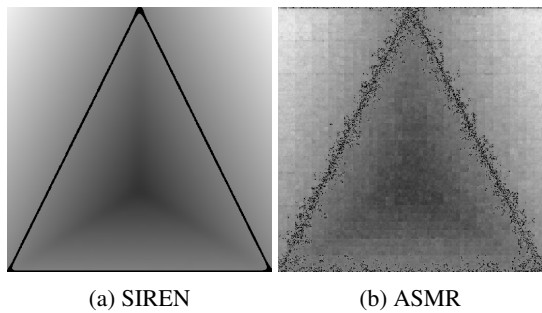

| (a) SIREN | (b) ASMR |

Figure 11: Cross section of a simple pyramid object.

