# OpenReview forum: "ASMR: Activation-Sharing Multi-Resolution Coordinate Networks for Efficient Inference"
_ICLR.cc/2024/Conference — ICLR 2024 poster_

### Official Review · Reviewer_rQpt · 2023-10-27

**Soundness:** 2 fair
**Presentation:** 2 fair
**Contribution:** 3 good
**Rating:** 8
**Confidence:** 4

**Summary:**

This paper proposes ASMR, a new architecture for implicit neural network (INR) representation. The main selling point is the very low inference cost. ASMR encodes a signal by creating a hierarchical partitioning of the input domain and associating to each partition level (except the first) one transformation layer and one modulation layer.  More precisely, each input dimension of size $N$ is partitioned in $L$ levels such that each level $x_i$ is of a given size $B_i$ and the product of all level sizes correspond to the size of the original dimension.  Every level $i$ of the model takes as input coordinate of that level ($x_i$), transforms it through a neural layer (the "modulator"), upscales both a linear transformation of the output of the previous layer and the output of the modulator, sums them and applies a non-linear function. This leads to reduced computation at inference: at a given layer $i\in\{1,...,L\}$ of the model, the number of computed values is multiplied by $B_i$ (for each dimension). In other words, the number of computed values increases from $B_0$ to $N$ with the depth of the model and values computed in the earliest layers are reused gracefully for many coordinates.

The paper then provides a proof that this architecture leads to an inference cost that is upper bounded by a value that does not depend on the depth of the network. Finally, experiments measures the inference cost and the reconstruction quality on images (Pluto and Kodak) and one video. Meta-learning and the impact of various strategy for the hierarchical decomposition are also evaluated and discussion in two distinct experiments. Between 2 and 5 baselines are considered in most experiments.

**Strengths:**

originality:
- As far as I know, the proposed approach is novel.

quality:
- The numerical experiments support the claim that ASMR inference cost does not depend on the number of layers.
- Inference cost is measured in MACs and not in running time.
- Despite the lower inference cost, representation quality is similar to or better than the baselines used.
- Comparison to the most relevant approaches kilo-NERF and LoE.

clarity:
- The paper is reasonably well written and clear.
- Experiments are well explained.

significance:
- Inference cost for INR is imho an important and understudied problem.
- The results seem very good.

**Weaknesses:**

quality:
- While several baselines are used in all experiments, instant-NGP is not included in the experiments measuring inference cost. As Instant-NGP is known for its fast inference, I think it is not possible to know whether ASMR is really better than grid-based approaches.
- In Figure 2, left, kiloNERF always has a higher MAC count than ASMR. In Figure 2, right, kiloNERF often has a lower MAC than ASMR. From the text, I assumed these were the same trained models. This looks inconsistent to me.
- In Section 3.3, the paper states that the per-sample inference cost depends only on the width of the model and is independent of the depth. I am no sure I agree with that statement, as the depth depends on the choice of the basis and thus on the width of the layers. For example, as far as I understand, doubling the size of the basis also means halving the number of layers.

clarity:
- The paper did not look polished in some places to me. For example, I am under the impression that multiple notations are used to refer to the same or very similar things, which was confusing to me. Here are a few examples:
   - a base is denoted by $|x_i$ in Figure 1 and $B_i$ in Section 3.1 (where Figure 1 is referenced)
   - the cumulative product of bases is denoted by $|x_{i...0}|$ in Figure 1 and by $C_i$ in Section 3.1.

significance:
- For the video experiment, a single video is used. I find that a bit underwhelming as recent papers have typically used video benchmarks for evaluation.

**Questions:**

I would be grateful to the authors for commenting on the weaknesses listed above.

Details/suggestions:
- Upon my first reading of the paper, I had some trouble understanding Figure 1. I think it would be helpful, but not necessary, to mention that all bases are of size 2.
- There is a problem with equation 1: Section 3.3: Equation 1 can be rewritten as [mathematical expression] (1).

Typos:
- Figure 1, b: hierarcical

---

> ### Author Response · Authors · 2023-11-18
> **Response to Reviewer rQpt**
>
> Thank you for the valuable comments and feedback.
>
> **Comparison to Grid-based Methods**
>
> We are aware of grid-based INRs such as Instant-NGP (Megapixel image, SDF, NeRF) and NVP (Video) that have demonstrated excellent training and inference efficiencies. However, just like [3] stated, grid-based methods come with the trade-off of losing the ability to learn global latent representations of data. Intuitively this is because features learned by grid-based methods are highly concentrated on the explicit grid instead of the implicit network, which is overly position-dependent. This takes away the INR's ability to learn abstracted features of the data and hence jeopardizes INRs' potential to be an implicit representation for downstream tasks or an encoder for an entire dataset. Indeed, it was our intention to include Section 5.2 for empirically demonstrating this on a meta-learning task to encode an entire CIFAR-10 dataset.
>
> As such, we believe that grid-based INRs and purely implicit INRs should be seen as complementary representations where direct comparison is not applicable. The main focus of our work is not to beat the state-of-the-art INRs for specific modalities, but rather to propose a generic technique for reducing the inference cost of purely implicit INRs *without* sacrificing their original reconstruction capabilities.
>
> **Clarification on Figure 2**
>
> We apologize for the unclear description and appreciate you pointing this out. The left and right figures are meant to show different sets of KiloNeRF and LoE model configurations. The intention of the left figure was to complement Table 1 (analytical relationship between MAC, \#Params, depth) with empirical data. The model configurations are chosen such that all the parameter counts are within the same range across the 4 models and the correlations between MAC and \#Params can be clearly demonstrated. On the other hand, the right figure is an extension of Table 2 (ultra-low MAC fitting results) showing the full MAC-PSNR curves of the models when having 3 to 6 layers. In our revised manuscript, we have separated the two figures and positioned them alongside the tables that they are meant to complement. Additional model configurations of the ultra-low MAC experiments are added to Appendix B accordingly.
>
> **Decoupling Between MACs and Depth**
>
> By width, we are referring to the number of hidden units in a network, which is independent of the choice of basis. If you are referring width to the number of unique inputs (coordinates) that each layer of the network needs to infer, then you are correct in that "doubling the size of the basis" would halve the number of layers *and* increase the width (number of unique inputs) by 2. However, as provided in the original manuscript, the derivation of ASMR's inference MAC is as follows:
> \begin{align*}
>     \text{MAC} = \sum_{i=1}^LB^iM_i \leq M\sum_{i=1}^LB^i &= MB\frac{B^L - 1}{B - 1} \approx \frac{MB}{B-1}B^L = \frac{MB}{B-1}B^{\log_B N} = \frac{B}{B-1}MN \leq 2MN
> \end{align*}
>
> where the MAC is bounded above by $2MN$ regardless of the choice of basis since $\frac{B}{B-1} < 2$ for all $B$. Hence, asymptotically, ASMR does indeed decouple its inference cost from the network depth. In fact, you may see from the right figure of Figure 2 in the original manuscript that the MAC of ASMR is not strictly but *approximately* constant in a practical setting.
>
>
> **Additional Video Results**
>
> We have tested ASMR on the UVG dataset, which is commonly used as a benchmark for testing video representations such as in the works of [1] and [2]. The results are consistent with our findings on the cat video presented in the original manuscript. Please refer to our General Response which contains additional experimental results on audio and 3D shapes.
>
>
>
> **Writing Issues**
>
> We have made the following changes (highlighted in red) in the revised manuscript:
> - We fixed the typo and specified that a uniform base of 2 is used in the caption of Figure 1.
> - We replaced the notations of base and cumulative base in Figure 1 so that they are in line with those defined in Section 3.1.
> - We fixed the labeling issue of Equation 1.
>
> **References**
>
> [1] Adversarial Generation of Continuous Images, CVPR 2021
>
> [2] Generating Videos with Dynamics-aware Implicit Generative Adversarial Networks, ICLR 2022
>
> [3] Implicit Neural Representations with Levels-of-Experts, NeurIPS 2022

---

> > ### Comment · Reviewer_rQpt · 2023-11-22
> > **Thank you for your answer**
> >
> > **Comparison to Grid-based Methods**
> > You make a good argument. Thank you for pointing this out.
> >
> > **Clarification on Figure 2**
> > This looks much clearer, I think this is a great update.
> >
> > **Decoupling Between MACs and Depth**
> > Indeed I did mean the number of hidden units in a layer. Sorry if this what not clear. I agree the proof is correct and that the computational cost is bounded and the empirical results back that up. I also agree that the inference cost is asymptotically  decoupled from the network depth.
> >
> > To clarify, my issue is more with the wording of these elements in the text. More precisely:
> > - The text states that "the inference cost of ASMR dependents only on the width of the model (i.e. number of hidden units) and independent of the depth". The width and the depth of the ASMR architecture are correlated as the depth and width are constrained by the dimension $N$ and the choice of the basis. So imho it is impossible to be dependent (or independent) on either the width or the depth and not the other.
> > - Proposition 1 states that "ASMR decouples its inference cost (in terms of MAC) from its depth L". The proof shows that you can upper bound the inference cost by 2 MN, which does not depend on the depth L. I agree with that. For me, that is not the same as proving that the inference cost is decoupled from the inference cost.
> >
> > I think adding talking about of "maximum" inference cost or "asymptotic" inference cost as in your answer would make things correct.
> >
> > **Additional Video Results**
> > Thank you for the additional results on UVG. They look good.
> >
> > I think I will update my score and may update them further while discussing with other reviewers. Like reviewer 24bM , I would be inclined to give a 7.

---

> > > ### Author Response · Authors · 2023-11-23
> > > **Thank you for your comment and clarifications on the "decoupling" statement**
> > >
> > > We are pleased to hear that our additional experiments and amendments to the manuscript are well perceived. We would also like to express our sincere gratitude for your thoughtful comments and criticisms. It has brought fundamental improvements to our work.
> > >
> > > We agree that adding "asymptotic" to Proposition 1 would make the statement more accurate. However, may you clarify why you believe that:
> > > >The width and the depth of the ASMR architecture are correlated as the depth and width are constrained by the dimension and the choice of the basis
> > >
> > > Under the assumption that parameters are NOT held fixed, we don't see a correlation between the # of hidden units and the choice of basis (hence network depth). Instead, the # of hidden units is a fully flexible hyperparameter that could be chosen arbitrarily by the user. We suspect that this confusion may be due to Figure 1(c) in the manuscript. We clarify by stating that the square in Figure 1(c) represents the feature maps of the entire image, and each "pixel" in the square is the feature generated by a forward pass of a **single** coordinate through the modulator, with length determined by the # of hidden units. That is, the choice of basis only affects the size of the square but NOT the dimension of the features

---

### Official Review · Reviewer_24bM · 2023-11-01

**Soundness:** 4 excellent
**Presentation:** 2 fair
**Contribution:** 3 good
**Rating:** 6
**Confidence:** 4

**Summary:**

This work (ASMR), studies a way of reducing the inference cost of coordinate networks. ASMR combines multi-resolution coordinate decomposition with hierarchical modulations, by sharing activations across grids of the data which decouples the inference cost from the depth. Comparisons against the popular SIREN model are given which shows ASMR outperforms SIREN in both computing cost and reconstruction performance.

**Strengths:**

1. The pipeline is elegant and seems not difficult to implement. The preconstruction metrics are at least on par with SIREN while the theoretical computation cost is vastly reduced.

**Weaknesses:**

W1. The number of MACs is certainly a good indicator of inference speed. It would be much more convincing to also have actual inference wall clock timing on several hardware platforms.

W2. It is okay to test only on image/video fitting tasks, but a study on other data types such as audio or 3D shape could be much more convincing to show the generality of the multi-resolution activation-sharing schema.

W3. The word ‘inference bandwidth’ is kinda misused. Maybe consider ‘throughput’ or ‘latency’.

Typo:

W4. Section 3.3 “Equation 1 could be rewritten as”. The original equation 1 was not labeled.

**Questions:**

1. How were the MACs counted (in terms of software implementation)?
2. My understanding from Figure 1. c about the modulator M is it adds the same pattern details to coarser coordinated activations (upsampled). It is counterintuitive why the sample pattern beats SIREN. Could the authors provide some explanations or intuitions?

---

> ### Author Response · Authors · 2023-11-18
> **Response to Reviewer 24bM**
>
> Thank you for your comments. We appreciate your positive feedback and recognition of our work.
>
> **Latency**
>
> To demonstrate the improved inference speed of our ASMR method on resource-limited devices, we measured the rendering latency of a 512 $\times$ 768 image (same as Kodak) on various hardware platforms. These platforms range from high-end to low-end CPUs, including a server-level AMD 7413 2.65GHz 24-Core CPU, a desktop 3.1 GHz Dual-Core Intel Core i5 CPU, and a Quad-core Cortex-A72 (ARM v8) 1.8GHz CPU of Raspberry Pi 4 model B. The reported latency is an average across 10 runs. The instantiation of each model is the same as the one mentioned in Section 4.3 of the manuscript, with parameter counts of around 130K. It can be observed that ASMR consistently speeds up SIREN by $>3\times$ in terms of latency, and the speed-up becomes more significant when tested on devices with reduced computation power (viz. $>4\times$ on Raspberry Pi). Processes that are too computationally intensive and result in hanging or being automatically killed due to excessive RAM usage are indicated by (Failed).
>
> | Latency (ms) | FFN    | SIREN   | MFN      | WIRE     | ASMR   |
> |--------------|--------|---------|----------|----------|--------|
> | Server       | 369.5  | 441.4   | 459.9    | 2296.5   | 143.9  |
> | Desktop      | 2859.2 | 4400.0  | 10291.4  | (Failed) | 957.7  |
> | Raspberry Pi | 8975.4 | 13476.6 | (Failed) | (Failed) | 3202.0 |
>
>
>
> **Other Data Modalities**
>
> To further demonstrate ASMR's ability to generalize across multiple data modalities, we have tested on the Librispeed dataset (Audio), UVG dataset (Video), and ShapeNet dataset (3D shapes) and included the results in the general response. Consistent with our findings in the original manuscript, our new results show that ASMR successfully reduces the inference cost of a SIREN model without sacrificing (or sometimes brings an even better) reconstruction quality. In particular, we found that ASMR performs better with low-dimension data such as audio, with an exceptional improvement over vanilla SIREN. For more details, please kindly refer to the general response at the top.
>
>
>
> **MACs Calculation**
>
> DeepSpeed Profiler is sufficient in calculating the MACs of all the models except for ASMR. To obtain the per-sample MAC, we divide the MAC outputted by the `get_model_profile()` function by the size of the input. For ASMR, we first obtained the MACs of each individual layer and modulator using DeepSpeed Profiler, then manually amortized the per-layer inference cost across the grid by the resolution of the layer.
>
>
> **Clarification on Figure 1**
>
> Your understanding is correct, and this is exactly why ASMR could reduce the inference cost of SIREN by orders of magnitude. ASMR does not only achieve activation-sharing on the INR backbone but also on the modulators. We would like to highlight that even though the modulations also exhibit a repeated pattern, their pattern is actually different from that of the previous-layer activation (as indicated by different colors in Figure 1). The summation of both results in activations that are unique to each coordinate. Intuitively, one may think of modulations as a set of phase shifts to the sine activations generated by the SIREN backbone. ASMR does so efficiently by finding a common set of phase shifts that can be applied repeatedly to generate a unique combination of frequency supports representing the data.
>
> As for the superior representation performance of ASMR over SIREN, we argue that it is because SIREN suffers from spectral bias while ASMR does not. As illustrated in Appendix G, SIREN learns the lower frequency information first, then progressively adds higher frequency details. For data with abrupt changes in value (i.e. very high frequency) such as non-analytical 3D shapes, it may even fail to converge and get stuck with a "smooth'' representation. On the other hand, by injecting coordinates that repeat themselves (i.e. periodic) in progressively higher frequencies, ASMR could tackle a much wider band of frequencies from the start of the training. Its discretized nature also permits it to learn any sort of data as long as it is within its grid resolution.
>
>
>
> **Writing Issues**
>
> We have made the following changes (highlighted in red) in the revised manuscript:
>
> * We replaced "inference bandwidth" with "inference throughput"
> * We fixed the labeling issue of Equation 1.

---

> ### Comment · Reviewer_24bM · 2023-11-19
>
> Thank the authors for the experiments on actual inference speed and clarification on how the MACs were profiled. I am more convinced now of the claims in the work regarding the computational cost. So I increased the soundness score by 1 and and confidence score by 1.
>
> It's quite hard to raise the score from 6 to 8 and sadly due to ICLR's grading scale, it's not possible to give a 7. But looking at the additional experiments on audio and 3D shape and the reply to reviewer zbDF, I am inclined to a score slightly higher than 6. Although the results with video don't seem to be too promising (when the number of parameters is controlled). It is also worth detecting the reason behind this phenomenon.

---

> > ### Author Response · Authors · 2023-11-23
> > **Thank you for the comments**
> >
> > We are glad to hear that our additional experiments have improved the clarity and soundness of our work. Your feedback has been invaluable to us and we appreciate your careful review of our work.
> >
> > We will continue to investigate the reason behind our model's drop in performance in video tasks and explore our model's capabilities in other modalities such as neural radiance fields and sign distance fields. Although results may not come through before the official discussion period, we still plan to post updates here whenever appropriate and look forward to future discussions.

---

### Official Review · Reviewer_zbDF · 2023-11-06

**Soundness:** 3 good
**Presentation:** 3 good
**Contribution:** 2 fair
**Rating:** 5
**Confidence:** 5

**Summary:**

This paper tackles the inefficient inference of INRs. As INR requires computing the individual coordinate's value on all layers, the increase in depth largely affects the inference efficiency, which is a serious problem for high-resolution signals. To reduce the computation, this paper suggests sharing the activation by incorporating multi-resolution coordinates and hierarchical modulation. To be specific, these multi-resolution coordinates are shares the activation and use up-sampling to save the activation storage. The authors demonstrate the efficiency of the proposed method on image and high-resolution signals (e.g., extreme high-resolution image and video).

**Strengths:**

The overall writing is clear and the paper is well-organized.

Tackles an important problem, i.e., heavy inference time (cost) of existing INRs.

The overall idea is sounds and reasonable to reduce the inference cost of INRs.

**Weaknesses:**

**Missing important baseline/related work.**
- The overall concept is highly similar (although the detail is different) to [1].
- Using multi-scale INR is explored in [1] (also, [2] uses the same idea in the video domain). Use upsampling (bilinear, nearest, etc) to reduce the number of forward passes by sharing the activations.
- [1] also uses modulation as well.
- Due to the existence of [1], I think the paper should change the overall claim, e.g., the first INR to decouple MAC from its depth or using shared activations.
- Also, the comparison with Multi-scale INR [1] is definitely needed.

**Missing related works: INR modulation.**
- There exist more recent modulation techniques in the field compared to bias modulation [3,4]. It would also be interesting if ASMR uses more advanced modulations, e.g., low-rank modulation for SIREN [3].
- missing a reference [9], which is the same method as COIN++ but used for other downstream tasks (e.g., generation, classification).

**The method is somewhat non-trivial to use for 3D scene rendering**
- Note that 3D is one of the most famous applications of INR.
- For 3D INRs, there is some paper that tackles a similar issue [5].

**The experiment section should be improved**\
(i) Recently, there have been several papers that tackle modality-specific INRs. The authors should claim the benefit of using ASMR compared to modality-specific ones.
- First, the authors should show multiple modalities rather than images. (Note that video only shows one example). Considering Audio or Manifold [8]
- Second, the authors need to discuss the benefit of using ASMR over modality-specific INRs. For instance, there are several video INR papers [6,7] showing efficiency.
- Finally, I think ASMR is hard to use for 3D scenes, so it has less benefit compared to SIREN or FFN.

(ii) Only used one video sample for evaluation.
- Rather than reporting one sample, it is better to consider a dataset.

(iii) It is hard to understand the intention of Section 5.2 (as it is somewhat trivial).
- I think the authors are trying to point out that.
- (a) ‘ASMR uses modulation, but there is no specific modulation technique for multi-grid based INRs’
- (b) ‘meta-learning modulation is important when using INRs for downstream tasks due to the dimensional efficiency.’ [8,9]
- I think this claim is not new to the community, and the modulation technique that ASMR is using is from [8,9].

(iv) Comparison with grid-based baselines are missing, e.g., Instant-NGP [10].
- Instant-NGP shows image, high-resolution image, and video experiments.
- It is very worth comparing the efficiency as these methods are proposed for efficiency.

**Summary**\
Overall, I quite like the method, but there are several things to be improved, e.g., the major claim should be changed due to the missing important baseline, and comparing with grid-based INRs like instant-NGP [10] and Multi-scale INR [1]. I kindly request the authors to rewrite certain parts and compare ASMR with the baseline in detail throughout the rebuttal.

**Reference**\
[1] Adversarial Generation of Continuous Images, CVPR 2021\
[2] Generating Videos with Dynamics-aware Implicit Generative Adversarial Networks, ICLR 2022\
[3] Modality-Agnostic Variational Compression of Implicit Neural Representations, ICML 2023\
[4] Versatile Neural Processes for Learning Implicit Neural Representations, ICLR 2023\
[5] Mip-NeRF: A Multiscale Representation for Anti-Aliasing Neural Radiance Fields, ICCV 2021\
[6] NeRV: Neural Representations for Videos, NeurIPS 2021\
[7] Scalable Neural Video Representations with Learnable Positional Features, NeurIPS 2022\
[8] COIN++: Neural Compression Across Modalities, TMLR 2022\
[9] From data to functa: Your data point is a function and you can treat it like one, ICML 2022\
[10] Instant Neural Graphics Primitives with a Multiresolution Hash Encoding, SIGGRAPH 2022

**Questions:**

Please refer to the weakness section.

---

> ### Author Response · Authors · 2023-11-18
> **Response to Reviewer zbDF (Part I - Comparison to [1])**
>
> Thank you for the comments and suggestions.
>
> **Comparison to Multi-Scale INR [1]**
>
> In particular, we appreciate reviewer zbDF for drawing the relation between ASMR and [1] for the similarity in ideas. In this regard, we would like to emphasize the major improvements of ASMR over [1]. To demonstrate these improvements, we conduct a thorough ablation study on the Cameraman dataset, contrasting ASMR and [1]. Specifically, we analyze the following key differences: (1) Multi-Scale Coordinates vs. Coordinate Decomposition, and (2) FMM vs. Bias-only Modulation. Each model consists of 9 layers with a hidden size of 64 and uses a base 2 partition. All of them are trained for 10K iterations.
>
>
> | Method              | Coordinate Representation | Modulation | \#Params (K) | MACs (K) | PSNR (dB)$\uparrow$ | SSIM $\uparrow$ |
> |---------------------|---------------------------|------------|--------------|----------|---------------------|-----------------|
> | Multi-scale INR [1] | Multi-scale               | FMM        | 49.6         | 59.623   | 26.73               | 0.746           |
> | Ablation \#1         | Multi-scale               | Bias-only  | 30.4         | 1.600    | 16.21               | 0.148           |
> | Ablation \#2         | Coord. decomp.            | FMM        | 49.6         | 1.434    | 29.65               | 0.784           |
> | ASMR                | Coord. decomp.            | Bias-only  | 30.4         | 1.429    | 29.71               | 0.795           |
>
> We believe that there are two major differences between ASMR and [1]
>
> **1. Multi-Scale Coordinates vs. Coordinate Decomposition**
>
> ASMR is motivated by the observation that data often exhibits both hierarchical AND periodic patterns. As a result, ASMR chooses to decompose coordinates like a change of basis operation. This is contrary to [1] which simply uses progressively finer coordinates. As a consequence, ASMR only injects $B_i^d$ unique coordinates into a layer $i$, where $B_i$ is the basis of partition of layer $i$ (assume $B_i$ is uniform across dimensions) and $d$ is the dimension of the data, while multi-scale INR injects $C_i^d$ unique coordinates, where $C_i$ is the resolution (or equivalently cumulative basis) of layer $i$. Note that $C_i$ grows with each layer while $B_i$ is a predefined constant and $B_i \ll C_i$ in later layers. Even though activation-sharing is also employed in [1], the inference cost of an INR with multi-scale coordinates is not truly decoupled from its depth, as the modulation layers need to operate on increasing resolution (i.e. increasing $C_i$). Hence, when the full resolution is large, multi-scale INR will incur a noticeable overhead for calculating modulation while ASMR's forward pass of the modulator maintains a negligible constant, regardless of the choice of modulation as shown in the table above. In addition to reducing the number of MACs, the use of coordinate decomposition in ASMR introduces an additional inductive bias that takes into account of underlying periodic structure of data, alleviating the spectral bias that is often encountered by SIREN as illustrated in Appendix G. Our ablation study shown above demonstrates that ASMR's coordinate decomposition significantly enhances performance compared to multi-scale coordinates as well as reducing MACs.
>
> **2. FMM vs. Bias-only Modulation**
>
> Although ASMR and [1] both use modulation techniques, they have different motivations and objectives. The Factorized Multiplicative Modulation (FMM) proposed in [1] does not rely on positional information and only uses parameters predicted by the hypernetwork, which represents a specific style or image context. On the other hand, the hierarchical modulation proposed in ASMR is designed to easily incorporate decomposed coordinates using bias-only modulation that results in minimal overhead. One can implement FMM using multi-scale coordinates, similar to ASMR, where a modulator is used in each layer to generate positional-dependent modulation parameters. However, we argue that the multi-scale coordinate representation proposed in [1] is not efficient when combined with FMM. For instance, for an image with a resolution of $512^2$ (Cameraman dataset), the modulator of FMM in the final layer needs to operate on the full resolution. This means that the final FMM modulator has to modulate the corresponding weight matrix for each of the $512^2$ coordinates, introducing a significant overhead. In contrast, the modulator in ASMR only needs to operate on a small constant number of bases ($2^2$ when the base is 2), followed by an efficient upsampling operation. As shown in the above ablation study, AMSR, using the combination of coordinate decomposition and bias-only modulations, achieves the lowest MACs and the best reconstruction quality, significantly outperforming [1].
>
> **References**
>
> [1] Adversarial Generation of Continuous Images, CVPR 2021

---

> ### Author Response · Authors · 2023-11-18
> **Response to Reviewer zbDF (Part II - Other Related Works)**
>
> **Other Modulation Methods**
>
> We acknowledge that there exists a wide variety of methods for implementing modulation and have revised our manuscript by including the missing references (highlighted in red). However, we would like to highlight the motivation behind choosing bias modulation over other methods: its effectiveness and simplicity in terms of implementation and inference cost. We aim to introduce ASMR as a straightforward addition to a SIREN model, which can significantly reduce its inference cost without adding much complexity. Compared to advanced methods like variational modulation in [3,4] or low-rank modulation in [1], bias modulation is the best fit for these characteristics. Intuitively, bias modulation used in ASMR can be interpreted as phase shifts that describe the variation of decomposed coordinates at each hierarchical level. The above ablation study shows that bias-only modulation can effectively model such variation and even outperform the more complex low-rank modulation technique, which incurs more parameters.
>
> **Comparison to Modality Specific INRs**
>
> We fully agree that each modality has its own state-of-the-art (SOTA) that excels in efficiency and quality. However, we argue that most of these SOTAs are usually overly modality-specific in the sense that their tricks are specialized to excel only in the tasks that they are intended for. For instance, [1,2] take special measures for the time dimension and grid-based methods fail to learn global latent representations. In contrast, general backbone models like SIREN are applicable to all modalities and downstream tasks. Hence, we see great values in proposing ASMR as a simple, modularized addition to SIREN that reduces its inference cost while inheriting its generalization capabilities. The modularity of ASMR also makes it orthogonal to other INR techniques such that they could be potentially combined for specific modalities or tasks.
>
> **Comparison to Instant-NGP and Intention of Section 5.2**
>
> We are aware of grid-based INRs such as Instant-NGP (Megapixel image, SDF, NeRF) and NVP (Video) that have demonstrated excellent training and inference efficiencies. However, just like [5] stated, grid-based methods come with the trade-off of losing the ability to learn global latent representations of data. Intuitively this is because features learned by grid-based methods are highly concentrated on the explicit grid instead of the implicit network, which is overly position-dependent. This takes away the INR's ability to learn abstracted features of the data and hence jeopardizes INRs' potential to be an implicit representation for downstream tasks or an encoder for an entire dataset. Indeed, it was our intention to include Section 5.2 for empirically demonstrating this on a meta-learning task to encode an entire CIFAR-10 dataset.
>
> As such, we believe that grid-based INRs and purely implicit INRs should be seen as complementary representations where a direct comparison is not applicable. The main focus of our work is not to beat the SOTA INRs for specific modalities, but rather to propose a generic technique for reducing the inference cost of purely implicit INRs *without* sacrificing their original reconstruction capabilities.
>
> **References**
>
> [1] Adversarial Generation of Continuous Images, CVPR 2021
>
> [2] Generating Videos with Dynamics-aware Implicit Generative Adversarial Networks, ICLR 2022
>
> [3] Modality-Agnostic Variational Compression of Implicit Neural Representations, ICML 2023
>
> [4] Versatile Neural Processes for Learning Implicit Neural Representations, ICLR 2023
>
> [5] Implicit Neural Representations with Levels-of-Experts, NeurIPS 2022

---

> ### Author Response · Authors · 2023-11-18
> **Response to Reviewer zbDF (Part III - Experiments Related)**
>
> **Other Data Modalities: Audio and 3D Shapes**
>
> To demonstrate the versatility of ASMR in handling different modalities, we present additional experimental results on audio and 3D shape datasets (please kindly refer to the general response at the top). We found that ASMR performs exceptionally well in particular on low-dimensional data like audio. Not only does it exhibit low MACs behavior, but it also leads to faster convergence and remarkably better results vs vanilla SIREN.
>
> It is a valid concern that ASMR's discrete nature may make it difficult to extend to 3D data. However, we argue that as long as the data is discretizable (which is the case for almost any data including SDFs and neural fields), ASMR could effectively become a low-inference cost counterpart to a vanilla SIREN model. For instance, we trained an ASMR on 3D occupancy grids of random objects in ShapeNet, and found that matches the quality of SIREN while operating at a significantly lower inference cost (for more details, please refer to the general response at the top).
>
>
> **Additional Video Experiments**
>
> We present additional results for the video-fitting task on the UVG dataset, which is a commonly used benchmark for testing video representations. The results are consistent with our findings on the cat video presented in the original manuscript. For details please refer to our general response at the top.

---

> ### Author Response · Authors · 2023-11-23
> **Thank you and follow-up discussion**
>
> We would like to express our sincere appreciation for your detailed review posted previously. From both our own perspective and that of reviewer 24bm and rQpt, the additional experiments and comparisons to baselines have significantly improved our work.
>
> As the discussion deadline is approaching, we kindly request your feedback on whether the follow-up response adequately addresses your concerns. Your experienced comments on the subject matter would be invaluable to our work's future explorations.
>
> Your timely response is greatly appreciated.
>
> Thanks.

---

> ### Comment · Reviewer_zbDF · 2023-11-23
> **Thank you for the response**
>
> Sorry for the late reply. Thank you very much for the reply. However, I still have the following concerns. I believe most of them can be address by editing.
>
> --------
>
> First, comparing with modality-specific INRs is for addressing and discussing the limitations of the method and does not hurt the paper's contribution. Rather, it is good for the readers, and it is a benefit to the paper.
> - Furthermore, I do agree that this work can be used for 3D-SDF. But is it applicable for 3D rendering? If it is not, discussing it in the limitation section will be great (since 3D rendering is one of the most important applications in INRs). I don't think this is not a negative point or hurts the paper's contribution. Addressing it in the limitation or discussion section will be much better and even strengthen the paper.
>
> --------
>
> Second, I still believe that we need a comparison with Instant-NGP or give a better reason that we need efficient MLP-based INRs (I have tried to address this reason in the paragraph below).
> - It is hard to understand the intention of the following sentence, “Intuitively, this is because features learned by grid-based methods are highly concentrated on the explicit grid instead of the implicit network, which is overly position-dependent.” Why is this a negative thing about grid-based INRs, and why should we use MLP-based INRs instead?
> - Also, the performance of meta-learning on Instance-NGP may be affected by the meta-learning method and setup. Furthermore, I still think the comparison on the meta-learning scenario does not give information to understand the benefit of this work. If one can compare instant-NGP on the meta-learning scenario, why do the authors not compare it with the main experiment (e.g., PSNR, MAC).
>
> For this paper, I think it is better to highlight the benefits of MLP-based INRs rather than show the meta-learning experiments. For instance, there are several works that use MLP-based INRs for downstream tasks [1,2,3,4,5]. I think the authors should claim this is the reason why the community should consider efficient MLP-based INRs (therefore, we don't need to compare with Grid-based INRs, since it is non-trivial to use Grad-based INRs for downstream tasks + MLP-based INRs need more efficiency for better downstream tasks).
>
> [1] From data to functa: Your data point is a function and you can treat it like one\
> [2] COIN++: Neural Compression Across Modalities\
> [3] Modality-Agnostic Variational Compression of Implicit Neural Representations\
> [4] HyperDiffusion: Generating Implicit Neural Fields with Weight-Space Diffusion\
> [5] Locality-Aware Generalizable Implicit Neural Representation
>
> --------
>
> Third, (sorry for the late additional question, this does not affect the score), I think there are several aspects of efficiency, including throughput, FLOPs, and training GPU days. I think the authors have well addressed the MAC (and throughput or Latency during the rebuttal), additionally, considering other metrics will be interesting as well. I think adding these metrics will be great if the paper gets accepted. Also, I think adding the Latency result in the paper would be much better.
>
> --------
>
> Finally, I believe this paper should include the results of multi-scale INR in the main paper and did not highlight enough. Since the main concept of the paper is to share the activation (as the method itself consists of this terminology), the paper should highlight that they are not the first to use this idea (only related work shows it). In the introduction, the reader still may get confused that this paper is the first to argue this concept as the activation sharing has no reference and is highlighted as bold.
>
> --------
>
> I think publishing this paper has good benefits for the community, but I feel that the paper should improve the paper by addressing the following.
> - discussing the limitations, highlighting the prior works, and why efficiency for MLP-based INR is needed even though we have efficient Grid-based INRs.

---

### Author Response · Authors · 2023-11-18
**General Response**

To showcase the versatility of ASMR in handling various modalities, we present additional experiment results on audio, video, and 3D shape datasets.

**Audio**

 Here, we evaluate ASMR performance on an audio-fitting task. Specifically, we use the LibriSpeech dataset as our benchmark. We select the first 100 samples from the test-clean split and train the models for 10K iterations using the first 2 seconds of each sample. All models consist of 4 layers with 128 hidden units. The average PSNR $\pm$ standard deviation is reported. From the following table, we can see that ASMR, with  $>30\times$ MAC reduction, significantly outperforms the baseline SIREN with an absolute gain of over 14dB.

| Method | MACs (K) | #Params (K) | PSNR (dB) $\uparrow$ |
|--------|----------|-------------|------------|
| SIREN  | 33.0     | 33.4        | 46.09±3.30 |
| ASMR   | 1.0      | 33.8        | 60.43±1.66 |


**Video**

We also test ASMR on the UVG dataset, a common benchmark used for evaluating video representations which contains 7 HD videos captured at 120fps. Due to time and hardware constraints, we downsample the videos by a factor of 4 to 150 frames of 270 $\times$ 480 resolution so that a model can reach decent reconstruction quality while fitting into a single NVIDIA 3090 GPU. We train a 10-layer SIREN with 512 hidden units, its ASMR counterpart, and a 10-layer SIREN (denoted as SIREN-S) with 128 hidden units with the same MAC order as the ASMR. Our results on the UVG dataset match our original observations on the cat video. We also stress here that at similar MACs, ASMR substantially outperforms its SIREN counterpart in terms of performance performance.


| Method  | MACs (M) | #Params (M) | PSNR (dB) $\uparrow$|   SSIM $\uparrow$  |
|---------|----------|-------------|-----------|------------|
| SIREN   | 2.10     | 2.10        | 35.2±3.95 | 0.921±0.04 |
| SIREN-S | 0.13     | 0.13        | 26.6±3.95 | 0.710±0.12 |
| ASMR    | 0.12     | 2.12        | 33.2±4.71 | 0.873±0.07 |


**3D Shape**

We demonstrate the direct extension of ASMR to 3D data by training it to fit $1024^3$ occupancy grids. We pick one random sample from each category of ShapeNet for a total of 55 objects and evaluate them with intersection-over-union (IoU). We train a 10-layer SIREN with 512 hidden units and its ASMR counterpart with a basis of partition of 2 for 200 epochs. All hyperparameters are kept constant across the two models. With a $>50\times$ MAC reduction by ASMR, we even observe a slight increase in IoU.


| Method | MACs(M) | #Params(M) | IoU $\uparrow$|
|--------|---------|------------|------------|
| SIREN  | 2.100   | 2.10       | 89.27±6.43 |
| ASMR   | 0.039   | 2.12       | 89.57±6.72 |

Consequently, these newly added experimental results, spanning multimodalities from audio, video to 3D shape, have further showcased the efficacy and superiority of ASMR, especially in its massive reduction in MAC counts while upkeeping performance. In fact, we are not aware of any INR works pinpointing order(s) of MAC reduction from an architectural perspective like ours, and we trust ASMR will definitely contribute a new and unique insight on low-cost INR inference. With the additional proof presented herein on the merits of ASMR, we are hopeful the reviewers will reevaluate our paper more favorably.

---

### Meta-Review · Area_Chair_A2ew · 2023-12-08

**Metareview:**

This paper tackles the problem of inference inefficiency for Neural Fields, where individual coordinates are typically treated as separate data points, thus requiring separate forward passes, a property that quickly becomes prohibitive for high-resolution signals. Instead, the authors propose the use of multi-resolution coordinate decomposition and hierarchical modulations, largely decoupling inference cost from network depth. Experiments show a comparison between inference complexity and achieved reconstruction performance.

Strengths:
- Reviewers agree that this is an important problem worth studying (zbDF, rQpt).
- The idea is sound and reasonable/elegant as well as easy to implement (zbDF, 24bM).
- Experimental results seem promising, especially considering the authors' rebuttal, having added: Librispeed dataset (Audio), UVG dataset (Video), and ShapeNet dataset (3D shapes) as well as a comparison with MultiScaleINR and latency results.

Weaknesses:
- Reviewer zbDF succinctly highlighted several points that remain to be improved, namely "discussing the limitations, highlighting the prior works, and why efficiency for MLP-based INR is needed even though we have efficient Grid-based INRs."

Decision:
- Given the inclination by two reviewers to raise their score (and a partial addressing of some of the concerns of the third reviewer), I now believe this paper should be accepted.
- However, as the paper would benefit from a reorganization and overhaul incorporating all responses made during the rebuttal, **I expect this to be done for the camera-ready version, including at a minimum an improved discussion of limitations and highlighting of prior works** and preferably a comparison with Instant-NGP (mentioned by two reviewers). As such, the authors should consider this to be a conditional acceptance with expected camera-ready changes. I trust this will be honored.

**Justification For Why Not Higher Score:**

No reviewer was ready to give a score of 8, having preferred a 7 on both occasions if available (see discussion).

**Justification For Why Not Lower Score:**

Both reviewers 24bM & rQpt indicated that they would have optimally liked to increase their score from 6->7 if possible, with one reviewer increasing their score to 8 and another leaving their score at 6. Given that all reviewers feel that the submission has improved during the rebuttal process, the fruitful discussion between authors and reviewers, and a clear path for improvement in the camera-ready version, I feel that the threshold for acceptance has been reached.

---

### Decision · Program_Chairs · 2024-01-16

Accept (poster)